# Rayleigh wind retrieval for the ALADIN airborne demonstrator of the Aeolus mission using simulated response calibration

Xiaochun Zhai[1,2], Uwe Marksteiner[1], Fabian Weiler[1], Christian Lemmerz[1], Oliver Lux[1], Benjamin Witschas[1], Oliver Reitebuch[1]

[1]Deutsches Zentrum für Luft- und Raumfahrt e.V. (DLR), Institut für Physik der Atmosphäre,

Oberpfaffenhofen 82234, Germany

[2]Ocean University of China, College of Information Science and Engineering, Ocean Remote Sensing Institute,

Qingdao 266100, China

*Correspondence to*: Oliver. Reitebuch (Oliver.Reitebuch@dlr.de)

**Abstract.** Aeolus, launched on August 22$^{nd}$ in 2018, is the first ever satellite to directly observe wind information from the surface up to 30 km on a global scale. An airborne prototype called ALADIN Airborne Demonstrator (A2D) was developed at the German Aerospace Centre (DLR) for validating the Aeolus measurement principle based on realistic atmospheric signals. To obtain accurate wind retrievals, the A2D uses a measured Rayleigh response calibration (MRRC) to calibrate its Rayleigh channel signals. However, differences exist between the respective atmospheric temperature profiles that are present during the conduction of the MRRC and the actual wind measurements. These differences are an important source of wind bias since the atmospheric temperature has a direct effect on the instrument response calibration. Furthermore, some experimental limitations and requirements need to be considered carefully to achieve a reliable MRRC. The atmospheric and instrumental variability thus currently limit the reliability and repeatability of a MRRC. In this paper, a procedure for a simulated Rayleigh response calibration (SRRC) is developed and presented in order to resolve these limitations of the A2D MRRC. At first the transmission functions of the A2D Rayleigh channel double-edge Fabry-Perot interferometers (FPIs) in the internal reference path and the atmospheric path are characterised and optimized based on measurements performed during different airborne and ground-based campaigns. The optimized FPI transmission functions are then combined with the laser reference spectrum and the temperature dependent molecular Rayleigh backscatter spectrum to derive an accurate A2D SRRC which can finally be implemented into the wind retrieval. Using dropsonde data as a reference, a statistical analysis based on dataset from a flight campaign in 2016 reveals a bias and a standard deviation of line-of-sight (LOS) wind speeds derived from a SRRC of only 0.05 m s$^{-1}$ and 2.52 m s$^{-1}$, respectively. Compared to the result derived from a MRRC with a bias of 0.23 m s$^{-1}$ and a standard deviation of 2.20 m s$^{-1}$, the accuracy improved while the precision is considered to be at the same level. Furthermore, it is shown that the SRRC allows the simulation of receiver responses over the whole altitude range from the aircraft down to sea level, thus overcoming limitations due to high ground elevation during the acquisition of an airborne instrument response calibrations.

# 1 Introduction

Continuous global wind observations are of highest priority for improving the accuracy of numerical weather prediction as well as for advancing our knowledge of atmospheric dynamics (Stoffelen et al., 2005; Weissmann et al., 2007; Žagar et al., 2008; Baker et al., 2014). Among various techniques such as radiosonde, radar wind profiler, and geostationary satellite imagery, a spaceborne Doppler wind lidar is considered as the most promising one to meet the need of near-real time observations of global wind information. Based on the principle of the Doppler effect, two different wind lidar detection techniques, namely coherent and direct detection, have been developed and studied over the last decades (Reitebuch, 2012a). The coherent Doppler lidar (CDL), typically used in the particle-rich boundary layer, can directly determine the Doppler frequency shift via the beat signal between the emitted laser signal and the particulate backscattered light, and the frequency shift introduced by an acoustic-optical modulator enables the measurement of positive and negative winds. In contrast, for a direct detection wind lidar the measured signal cannot directly be related to the frequency shift. Thus, a so-called response calibration describing the relationship between the measured instrument response and the actual Doppler frequency shift constitutes a prerequisite for an accurate wind retrieval. A direct detection wind lidar can measure atmospheric wind by means of either particulate or molecular backscatter signals, typically offering much higher data coverage of the wind field from ground up to the lower mesosphere. Different spectral discriminators such as Fabry-Perot interferometers (Chanin et al., 1989; Korb et al., 1992), Fizeau interferometers (McKay, 1998; McKay, 2002), iodine vapor filters (Liu et al., 2002; She et al., 2007; Baumgarten, 2010; Wang et al., 2010; Hildebrand et al., 2012), Michelson interferometers (Thuillier et al., 1991; Herbst et al., 2016) and Mach-Zehnder interferometers (Bruneau, 2001; Bruneau and Pelon, 2003; Tucker et al., 2018) can be used for direct detection wind lidars.

Aeolus, launched on August $22^{nd}$, 2018, is the first ever satellite to directly observe line-of-sight (LOS) wind profiles on a global scale. Its unique payload, the Atmospheric LAser Doppler INstrument (ALADIN), is a direct detection wind lidar operating at 355 nm from a 320 km orbit (Stoffelen et al., 2005; ESA, 2008; Reitebuch, 2012b). The backscatter signals from particulate and molecular backscatter are received by two different spectrometers, which are a Fizeau interferometer in the Mie channel, measuring particulate backscatter, and a double-edge filter with two Fabry-Perot interferometers (FPIs) in the Rayleigh channel, measuring molecular backscatter. The novel combination of these two techniques, integrated for the first time into a single wind lidar, expands the observable altitude range from ground to the lowermost 30 km of the atmosphere. ALADIN provides one component of the wind vector along the instrument LOS with a vertical resolution of 0.25 km to 2 km and with a requirement on the wind speed precision of 1 m s$^{-1}$ to 2.5 m s$^{-1}$ for the horizontally projected LOS (HLOS) depending on altitude. Furthermore, as the first high spectral resolution lidar in space (Ansmann et al., 2007; Flamant et al., 2008), ALADIN has the potential to globally monitor cloud and aerosol optical properties to contribute to the climate impact studies.

In the frame of the Aeolus program, a prototype instrument called ALADIN Airborne Demonstrator (A2D) was developed at the German Aerospace Centre (DLR). Due to its representative design and operating principle, the A2D has provided

valuable information on the validation of the measurement principle from real atmospheric signals before the satellite launch. In addition, the A2D is expected to contribute to the optimization of the wind measurement strategies for the satellite instrument as well as to the improvement of wind retrieval and quality control algorithms during satellite operation (Durand et al., 2006; Reitebuch et al., 2009; Paffrath et al., 2009). As the first ever airborne direct detection wind lidar, A2D has been deployed in several ground and airborne campaigns over the last 12 years (Li et al., 2010; Marksteiner, 2013; Weiler, 2017; Lux et al., 2018; Marksteiner et al., 2018).

Different instrument response calibration approaches have been studied using both measured and simulated response calibration to characterize and calibrate the ALADIN Rayleigh channel (Tan et al., 2008; Dabas et al., 2008; Rennie et al., 2017). Currently, only measured Rayleigh response calibrations (MRRC) are used for the A2D (Marksteiner, 2013; Lux et al., 2018; Marksteiner et al., 2018). However, the atmospheric temperature affects the Rayleigh-Brillouin line shape and has a direct effect on the instrument response calibration (Dabas et al., 2008). Differences exist between the respective atmospheric temperature profiles that are present during the conduction of the MRRC and the actual wind measurements. These differences are an important source of wind bias which grows with increasing temperature differences. This is also the reason why it is mandatory to consider the atmospheric temperature in the Aeolus level 2B procedure to retrieve reliable winds (Dabas et al., 2008; Rennie et al., 2017). Furthermore, some experimental limitations, which will be introduced specifically in Sect. 2.1, need to be considered carefully to achieve a reliable MRRC. Overall, the atmospheric and instrumental variability coming along with a MRRC limits the reliability and repeatability of A2D instrument response calibrations. Inspired by the calibration method used in the ALADIN level 2B processor (Dabas and Huber, 2017), the Simulated Rayleigh Response Calibration (SRRC) was developed to resolve these limitations of A2D. It is based on an accurate theoretical model of the FPI transmission function and the molecular Rayleigh backscatter spectrum. In this paper, the SRRC is introduced and its impact on the A2D wind retrieval is discussed and compared to results obtained with a measured response calibration.

In section 2, different calibration approaches of double-edge FPIs are discussed firstly. Afterwards, the principle of an A2D SRRC is presented in Section 3. Section 4 gives an overview over the campaign and the dataset analysed in this paper, whereas Section 5 introduces the A2D SRRC, which is applied to the campaign measurements, and discusses the corresponding wind results. Section 6 provides a statistical comparison of LOS wind velocities from A2D Rayleigh channel measurements using the MRRC and SRRC, and winds from simultaneous CDL and dropsonde datasets. A comparison of A2D MRRCs and SRRCs is also evaluated in Section 6. Section 7 provides a summary and conclusion.

## 2 Calibration approaches for double-edge FPIs

Chanin et al. (1989) demonstrated for the first time that FPIs can be used to measure wind in the middle atmosphere relying on molecular Rayleigh scattering and a laser with a wavelength of 532 nm. The so-called response can be defined as the contrast (Chanin et al., 1989) or the ratio (Korb et al., 1992) of the signal intensities obtained after transmission through the

FPIs. A response calibration is a prerequisite for wind retrieval since it represents the relationship between the measured quantity (e.g. intensity of the backscattered light) and the frequency shift which is induced by the Doppler effect. Generally, there are two approaches to determine the relationship between response and Doppler frequency shift, i.e. to obtain a response calibration function. Table 1 lists several FPI-based direct detection wind lidar systems that are capable of measuring wind information based on a measurement approach or a simulation approach.

For each direct detection wind lidar system, the emitted laser frequency should be known in order to allow an accurate derivation of the Doppler frequency shift. A zero Doppler shift reference determined by pointing to the zenith direction has been used to correct the short-term frequency drift in previous studies (Souprayen et al., 1999b; Korb et al., 1992; Dou et al., 2014). But for the A2D, the internal reference path is particularly dedicated to the derivation of information about the emitted laser frequency. As shown in Lux et al. (2018, Fig. 1), a small portion of the laser beam radiation is collected by an integrating sphere and coupled into a multi-mode fibre, then injected into the receiver via the front optics. This path is called internal reference path. The atmospheric backscattered signal is collected by a Cassegrain telescope and guided via free optical path propagation to the front optics and receiver successively. This path is called the atmospheric path. An electro-optical modulator is used to temporally separate the atmospheric signal from the internal reference signal, thereby avoiding disturbances of the internal reference signal by atmospheric signal and saturation of the detectors at short ranges (Reitebuch et al., 2009). Because of the different optical illumination of the internal and atmospheric path resulting in different divergence and incidence angles on the FPIs, the response calibration curves for these two paths are different. It is noted that the internal reference path of ALADIN is different from A2D's, where ALADIN uses free path propagation rather than fibre coupling unit.

## 2.1 Approach using measured response calibrations

The first approach to obtain a response calibration function is based on measurements during which the laser beam is pointed into zenith direction while assuming that the vertical velocity of the probed atmospheric volume is negligible, i.e. no Doppler frequency shift is induced. Then, either the frequency of the laser transmitter is scanned with constant FPIs cavity length (Reitebuch et al., 2018; Lux et al., 2018; Marksteiner et al., 2018) or the cavity length of the FPIs is scanned while keeping the laser frequency locked (Dou et al., 2014).

Since the shape of the actual molecular Rayleigh backscatter spectrum is determined by the atmospheric temperature and pressure profiles (Tenti et al., 1974; Pan et al., 2004), the measured response calibration function in the atmospheric path is only valid for a specific combination of temperature and pressure profiles. Regarding ground-based lidar systems, the calibration procedure can be carried out frequently. Based on stable atmospheric conditions (Dou et al., 2014; Liu et al., 2002) it is reasonable to assume that only small temperature and pressure variations occur with a negligible effect on the retrieved wind within a specific analysis period. However, for spaceborne or airborne lidar systems like ALADIN or the A2D, the

variability in temperature and pressure can be one of the main sources of systematic errors for the Rayleigh channel wind retrieval as it modifies the instrument response calibration (Dabas et al., 2008; Marksteiner, 2013).

For ALADIN, the Rayleigh winds produced by the level 1B processor (Reitebuch et al., 2018) are based on a MRRC while the level 2B processor uses a SRRC. A MRRC includes three response calibration curves, one each derived from the internal reference, the atmospheric and the ground return. A so-called instrument response calibration mode is usually performed once per week. During about 16 minutes the frequency of the laser transmitter is scanned over 1000 MHz in steps of 25 MHz and the satellite is rolled by 35° in order to point nadir, thereby avoiding frequency shifts induced by horizontal wind velocities. In order to increase the signal to noise ratio (SNR), the signals generally from the altitude range between 6 km and 20 km are accumulated to derive a single response calibration curve for the atmosphere (Reitebuch et al., 2018). Compared to ALADIN, the MRRC of the A2D can be derived and used per range-gate because of the larger SNR prevailing for airborne measurements which are performed closer to their target. The instrument response calibration of the A2D can be carried out several times during a flight by tuning the laser frequency in steps of 25 MHz over a frequency interval of 1.7 GHz.

Apart from the atmospheric temperature and pressure effect on the MRRC, several specific experimental constraints are critical for achieving a reliable instrument response calibration for both ALADIN and A2D. Firstly, the particulate Mie scattering which is not fully filtered out by the Fizeau interferometer will enter the FPIs and can be considered as Mie contamination of the Rayleigh signal. Because of the different spectral widths of the particle and molecular backscatter signal, the sensitivities of the FPIs on them are different. If not taken into account, the Mie contamination on the Rayleigh channel is one of the sources of systematic errors because it modifies the MRRC curve. In order to avoid such modifications, the A2D tries to conduct IRCs in preferably pure Rayleigh atmosphere. Furthermore, the characteristics of the ground, such as high albedo and preferably flat terrain as well as low ground elevation, should be considered to improve the SNR, to facilitate the deduction of a ground return response curve and to maximize the vertical coverage of the atmosphere (Marksteiner, 2013; Weiler, 2017; Lux et al., 2018; Marksteiner et al., 2018). In some cases, A2D calibrations were performed over terrain with high elevation (e.g. Greenland). Obviously, no response calibration curve can be obtained from below the surface, which would however be necessary for accurate wind retrieval at other geographical locations with lower ground elevation. In addition, the LOS velocity needs to be zero during the instrument response calibration. This is accomplished by flying curves with a roll angle of 20°, which corresponds to the installation angle of the A2D telescope in the DLR Falcon 20 aircraft. Regions showing gravity wave activity or strong convection are avoided as they cross the assumption of negligible vertical wind velocity (Lux et al., 2018; Marksteiner et al., 2018). Overall, the reliability and repeatability of A2D MRRCs is a main limitation for accurate wind retrieval.

## 2.2 Approach using simulated response calibrations

The second approach is based on SRRC curves and the fact that the transmitted signals through each FPI are proportional to the convolution of the respective filter transmission function with the atmospheric backscatter spectrum. Therefore, this approach relies on accurate models for both FPI transmission functions and atmospheric backscatter spectrum. In practice, the transmission function of FPIs can be obtained by scanning the laser frequency and keeping the FPI's etalon length fixed (Rennie et al., 2017) or scanning the spacing between the plates of FPIs with fixed laser frequency (Souprayen et al., 1999b; Xia et al., 2012).

For ALADIN and A2D, the seed laser is frequency tuneable to cover a spectral range of 11 GHz in the UV to calibrate the spectral characteristics of FPIs for the internal reference path. This procedure is called instrument spectral registration (Reitebuch et al., 2018). However, the transmission functions of FPIs for the atmospheric path are different from the transmission curves registered on the internal reference path during the instrument spectral registration. This is because of the difference in the illumination of the FPIs by the beams in the atmospheric and the internal reference paths, i.e. due to different divergence and incidence angles (Reitebuch et al., 2009). For ALADIN, this is taken into account by correcting the FPIs transmission curves of the atmospheric path (Dabas and Huber, 2017). Regarding the A2D, a SRRC based on such a simulation approach promises an improvement in terms of wind speed errors. A SRRC includes two response calibration curves derived from internal reference path and atmospheric path. The transmission function of the A2D FPIs in the internal reference path can be obtained during an instrument spectral registration. The determination of the transmission functions of the FPIs in the atmospheric path of the A2D is the most sophisticated part needed to accurately retrieve wind information by using a SRRC. Furthermore, FPI transmission functions should be a function of incidence angles, field of view, temperature, pressure, thickness, fitness and so forth. Regardless of measurement or simulation method, any angular alignment drift will change the incidence angles on the FPIs, resulting in a different transmission value. Referring to the Observatory of Haute Provence (OHP) Rayleigh lidar, the bias induced by instrument drifts can be eliminated based by a specific wind acquisition cycle strategy using the differences between vertical and titled position measurement (Souprayen et al., 1999a). For ALADIN or the A2D, the instrument drift is compensated by regularly performing instrument response calibrations and instrument spectral registrations on a weekly basis.

## 3 The principle of A2D SRRC

The Doppler frequency shift in LOS direction is derived from the difference between the frequency of the received atmospheric return $f_a$ and the emitted laser frequency $f_i$:

$$\Delta f = f_a - f_i \,, \tag{1}$$

The corresponding LOS velocity is derived from the Doppler shift equation using a laser wavelength of $\lambda_0$:

$$V_{LOS} = \frac{\lambda_0}{2}\Delta f \, , \tag{2}$$

In order to derive $f_i$ and $f_a$ from the A2D Rayleigh channel, the transmitted intensities $I_{A,B,INT}(f)$ and $I_{A,B,ATM}(f)$ through the FPI filters A and B are used for the internal reference path ($INT$) and the atmospheric ($ATM$) path, respectively:

$$I_{A,B,INT}(f_i) = \int_{-\infty}^{+\infty} T_{A,B,INT}(f)S_i(f_i - f)df \, , \tag{3}$$

$$I_{A,B,ATM}(f_a) = \int_{-\infty}^{+\infty} T_{A,B,ATM}(f)S_a(f_a - f)df \, , \tag{4}$$

$$S_a(f) = S_{RB}(f) + (\rho - 1)S_{mie}(f) \, , \tag{5}$$

Taking the transmitted intensity through filter A for instance, $I_{A,INT}(f_i)$ is the convolution of the filter A transmission function on the internal reference path ($T_{A,INT}(f)$) and the normalized laser reference spectrum $S_i(f)$ with the transmitted laser frequency $f_i$. Accordingly, $I_{A,ATM}(f_a)$ is the convolution of the filter A transmission function on the atmospheric path ($T_{A,ATM}(f)$) and the normalized atmospheric backscatter signal spectrum $S_a(f)$ with the centre frequency $f_a$. $S_a(f)$ consists of the broad molecular Rayleigh backscatter spectrum $S_{RB}(f)$ (the subscript $RB$ stands for Rayleigh–Brillouin) and the narrow particulate Mie backscatter spectrum $S_{mie}(f)$, as shown in Eq. (5). Here $\rho = 1 + \beta_{aer} / \beta_{mol}$ is the scattering ratio, where $\beta_{aer}$ and $\beta_{mol}$ are the particle and molecular backscatter coefficients, respectively.

As described by Garnier and Chanin (1992), the Rayleigh response is defined as:

$$R_x(f) = \frac{I_{A,x}(f) - I_{B,x}(f)}{I_{A,x}(f) + I_{B,x}(f)}, x = INT \text{ or } ATM \, , \tag{6}$$

where $x$ represents the case of the $INT$ or $ATM$ path.

In order to determine the $R_x(f)$ by means of Eqs. (3)-(6), accurate knowledge about $T_{A,B,INT}(f)$, $T_{A,B,ATM}(f)$, $S_i(f)$ and $S_a(f)$ is needed. Generally, the transmission function $T(f)$ of an ideal FPI can be expressed by the Airy function. However, small defects on the FPI mirror surfaces or imperfect illumination of the FPI could result in small deviations that have to be considered (McGill et al., 1998). It is shown that all these defects can be represented by a Gaussian defect term that modifies the model of the FPI transmission function $T(f)$ to (Witschas et al., 2012):

$$T(f) = \frac{1}{FSR}\left[1 + 2\sum_{k=1}^{\infty} R^k \cos\left(\frac{2\pi k f}{FSR}\right)\exp\left(-\frac{2\pi^2 k^2 \sigma_g^2}{FSR^2}\right)\right] \, , \tag{7}$$

where $FSR$ is the free spectral range of the corresponding FPI (A or B) on the respective measurement path ( $INT$ or $ATM$ ), $R$ is the mean reflectivity of the mirror surfaces and $\sigma_g$ is a defect parameter taking mirror defects into consideration.

The laser pulse line shape $S_i(f)$ , with its laser linewidth and emitted laser frequency, (Lux et al., 2018; Marksteiner et al., 2018) can be approximated by a Gaussian function. The spectral distribution of $S_{mie}(f)$ is similar to $S_i(f)$ as particles can be considered to cause no significant spectral broadening due to random motion. $S_{RB}(f)$ can be computed by using the Tenti S6 line shape model (Tenti et al., 1974; Pan et al., 2004) which has been widely applied in atmospheric applications. An easily calculated analytical expression of the Tenti S6 line shape model for atmospherically relevant temperatures and pressures is used herein (Witschas, 2011a, b; Witschas et al., 2014).

The measurement principle of the A2D Rayleigh channel signal is shown in Fig. 1 as an example for one frequency step during the instrument spectral calibration with no Doppler shift on the LOS. It is assumed that there is no Mie contamination on the Rayleigh channel in this case, that is, $\rho=1$ or $S_a(f)=S_{RB}(f)$ . $S_i(f)$ is depicted using a Gaussian function with a full width at half maximum (FWHM) of 50 MHz. $S_{RB}(f)$ is calculated for T=270 K and P=700 hPa. The transmitted integrated intensities of $S_a(f)$ through FPIs A and B, that is, $I_{A,ATM}$ and $I_{B,ATM}$ are indicated by light blue and light magenta filled areas, respectively.

The LOS wind velocity error $\Delta V_{MC}$ induced by Mie contamination is defined as the difference of the LOS wind velocities measured under purely atmospheric molecular conditions and conditions with a scattering ratio of $\rho$ . Figure 2 shows a simulation of $\Delta V_{MC}$ at T=223 K and P=301 hPa, where the x-axis and y-axis represent different response values and scattering ratios, respectively. Positive and negative $\Delta V_{MC}$ represent the overestimation and underestimation of the LOS velocity, respectively. An overestimation of LOS velocities occurs at response values less than 0.235 in this case. Larger scattering ratios result in larger overestimation, and the difference can get up to 13 m s$^{-1}$ in case of $\rho=3$ . According to previous studies (Dabas et al., 2008), the Mie contamination correction could improve the quality of Rayleigh winds in cases of intermediate $\rho$ , e.g. below 1.5. In this region the Mie signal is not high enough to guarantee an accurate Mie wind measurement but instead becomes rather significant for the Rayleigh channel (Sun et al., 2014; Lux et al., 2018). The value of $\rho$ , which is needed for the Mie contamination correction in the Rayleigh channel, is obtained by analysing the Mie channel signal. The detailed algorithm can be seen in (Flamant et al., 2017).

Following the procedure of the A2D instrument response calibration mode, the intensities transmitted through the FPIs and corresponding response values at each frequency step are calculated, eventually forming the SRRC of the internal reference path ( $R_{INT}(f)$ , blue line) and the atmospheric path ( $R_{ATM}(f)$ , black line) shown in Fig. 3 (a). It is noted that the procedure is

done assuming no Mie contamination in this case. The cross point frequency $f_c$ (red dotted line) in Fig. 3 (a) is derived from $R_{INT}(f)$ where $I_{A,i}(f) - I_{B,i}(f)$ is closest to zero (Marksteiner et al., 2018). The relative frequency $f'$ is defined as the difference between absolute frequency $f$ and $f_c$. Figure 3 (b) shows the simulated response functions $R_{INT}(f')$ and $R_{ATM}(f')$ within a relative frequency interval of $\pm 850$ MHz, where the interval corresponds to the area marked by the dashed red-square in Fig. 3 (a). A linear least-squares fit $R_{linearfit\_x}(f')$ is applied to the SRRC of the internal reference and atmospheric path, shown by the solid blue and black line in Fig. 3 (b). The linear fitting parameters sensitivity $\beta_x$ and intercept $\alpha_x$ are defined as:

$$\beta_x = \frac{\partial R_{linearfit\_x}(f')}{\partial f'}, x = INT \; or \; ATM, \tag{8}$$

$$\alpha_x = R_{linearfit\_x}(f' = 0) = R_{linearfit\_x}(f = f_c), \tag{9}$$

The non-linearity $\gamma_x(f')$ is defined as the difference between $R_x(f')$ and linear least-squares fit $R_{linearfit\_x}(f')$, that is, $\gamma_x(f') = R_x(f') - (\beta_x f' + \alpha_x)$. The different $\gamma_x(f')$ functions of the internal reference path and the atmospheric path are shown in Fig. 3 (c). For a wavelength of $\lambda_0 = 354.89$ nm, a LOS velocity of 1 m s$^{-1}$ translates into a frequency shift of 5.63 MHz . Taking a sensitivity $\beta_{ATM} = 5 \times 10^{-4}$ MHz$^{-1}$, the atmospheric non-linearity at -200 MHz almost reaches -0.02 which is equivalent to about -40 MHz, which in turn correspond to -7.1 m s$^{-1}$. Consequently, large errors in the derived LOS velocity would occur if $\gamma_x(f')$ is not taken into account. Therefore, a 5[th] order polynomial fit (Marksteiner, 2013; Lux et al., 2018; Marksteiner et al., 2018) is selected to model $\gamma_x(f')$, as shown in Fig. 3 (c) for $R_{INT}(f')$ ($R_{ATM}(f')$) as solid blue (black) line. A fit of the SRRC for the internal reference and atmospheric paths can be expressed as a sum of a linear fit and a 5[th] order polynomial fit:

$$R_{fit,x}(f') = \beta_x f' + \alpha_x + \gamma_x(f') = \beta_x f' + \alpha_x + \sum_{i=0}^{5} m_{i,x} f'^i$$

$$= (a_x + m_{0,x}) + (\beta_x + m_{1,x}) f' + m_{2,x} f'^2 + m_{3,x} f'^3 + m_{4,x} f'^4 + m_{5,x} f'^5 , \tag{10}$$

The difference between $R_x(f')$ and $R_{fit,x}(f')$ is defined as residual and shown in Fig. 3 (d) for the internal reference path (blue line) and the atmospheric path (black line), respectively. A periodic fluctuation can be seen but the maximum residual of the atmospheric path is less than $1.5 \times 10^{-4}$, corresponding to 0.053 m s$^{-1}$ for $\beta_{ATM} = 5 \times 10^{-4}$ MHz$^{-1}$. The absolute difference between the two residuals (INT-ATM) is even smaller.

## 4 Campaign and dataset

As part of the North Atlantic Waveguide and Downstream Experiment (NAWDEX) carried out in 2016 in Iceland, four aircraft equipped with diverse payloads were employed to investigate the influence of diabatic processes for midlatitude weather (Schäfler et al., 2018). The DLR Falcon 20 was deployed with the A2D and a well-established 2 $\mu$m CDL, offering an ideal
platform to demonstrate the capabilities of the A2D under complex dynamic conditions. A total of 14 research flights were performed with the Falcon aircraft during the NAWDEX campaign. The A2D was operated in wind measurement mode in most of the flight periods, while the instrument spectral registration mode was carried out on ground and during airborne measurements. Furthermore, two flights on September 28th 2016 and October 15th 2016 were carried out to obtain A2D instrument response calibrations. Six MRRCs have been performed in these two calibration flight periods. After comparison
and evaluation given by Lux et al. (2018), the 3rd calibration, which was carried out over an Iceland glacier on September 28th 2016 at 12:53 UTC, is chosen as the baseline of A2D Rayleigh wind retrieval, as it shows low Rayleigh residual errors and was not affected by clouds, instrument temperature drifts and outliers (Lux et al., 2018). The other three aircraft, that is, the German High Altitude and Long Range Research Aircraft (HALO), the French Service des Avions Français Instrumentés pour la Recherche en Environnement (SAFIRE) Falcon 20 and the British Facility for Airborne Atmospheric Measurements (FAAM)
BAe 146, were equipped with dropsonde dispensers to provide temperature, pressure, wind and humidity profiles (Schäfler et al., 2018). Time-space matching datasets between dropsonde and A2D can be used as both references to validate A2D wind measurements and to provide essential atmospheric temperature and pressure profiles for SRRC in this study. Table 2 provides an overview of datasets that are available from the 2016 flight campaign and are used for this study. It is noted that all matched dropsondes listed in Table 2 were dispensed from the HALO aircraft.

The transmission functions of the FPIs are reproducible, and the transmission characteristics are different for the internal reference and atmospheric path. The underlying difference in illumination includes both a difference in the spatial as well as in the angular distribution of the light. In particular, the use of a multimode fibre in the internal reference path gives rise to speckles, resulting in an intensity distribution which is markedly different from that of atmospheric path. As for the A2D instrument spectral registration during the NAWDEX campaign, the sampled transmission functions of the FPIs are obtained
from the internal reference only since the atmospheric return is convolved with a temperature dependent RB spectrum and the hard target ground return would be too variable due to albedo variation. The only sampled transmission functions of the FPIs from the A2D atmospheric path are available from the BRillouin scattering Atmospheric INvestigation on Schneefernerhaus (BRAINS) field campaign (Witschas, 2011c; Witschas et al., 2012), which was performed during Jan-Feb 2009 to demonstrate the effect of Brillouin scattering in real atmosphere. Unique to BRAINS was a horizontal pointing of the outgoing laser beam
in order to get a hard target return of a mountain with constant albedo in about 10 km distance. This allowed measurements of narrowband backscatter signal through the atmospheric path. The transmission functions of the FPIs were sampled by changing the laser frequency with steps of 50 MHz over a frequency range of 12 GHz with fixed FPIs. Here, different transmission

curves of FPIs from the BRAINS field campaign in 2009 and NAWDEX airborne campaign in 2016 will be used as candidate FPIs transmission curves for SRRC analysis.

## 5 Determination of the A2D response function and Rayleigh wind retrieval

A flowchart of the LOS wind velocity retrieval based on SRRC and MRRC is presented in Fig. 4. Firstly, the atmospheric temperature and pressure profiles are taken from dropsonde, radiosonde or model data to derive the atmospheric molecular backscattered spectrum using the analytical representation of Tenti S6 line shape model (Witschas, 2011a, b; Witschas et al., 2014). Then the transmission functions of FPIs are obtained by fitting the measured FPIs transmission characteristics based on Eq. (7). Afterwards the frequency scan of the laser transmitter during A2D instrument response calibration is simulated to derive the SRRCs for the internal reference and the atmospheric path. The measured response values $R_{ATM}$, $R_{INT}$ obtained from A2D wind velocity measurement mode are combined with the SRRC $R_{fit,ATM}(f^{'})$ and $R_{fit,INT}(f^{'})$. The Doppler frequency shift $\Delta f_{SRRC}$ due to LOS velocity is then derived from the difference of $f^{'}_{a,SRRC}$ and $f^{'}_{i,SRRC}$ (Reitebuch et al., 2018):

$$\Delta f_{SRRC} = f'_{a,SRRC} - f'_{i,SRRC} = \frac{R_{ATM} - \alpha_{ATM} - \gamma_{fit,ATM}(f'_{a,SRRC})}{\beta_{ATM}} - \frac{R_{INT} - \alpha_{INT} - \gamma_{fit,INT}(f'_{i,SRRC})}{\beta_{INT}}, \qquad (11)$$

The LOS velocity $V_{LOS,SRRC}$ is derived according to Eq. (2):

$$V_{LOS,SRRC} = \frac{\lambda_0}{2} \Delta f_{SRRC}, \qquad (12)$$

It is noted that LOS velocity herein includes not only the horizontal and a possible vertical wind component but also the contribution from the aircraft flight velocity. The correction of the flight-induced velocity $V_{LOS,aircraft}$ is calculated using the inertial navigation system and, GPS on-board the aircraft within an attitude correction algorithm (Marksteiner, 2013). Finally, the corrected LOS wind velocity $V_{cor,SRRC}$ is obtained as follows:

$$V_{cor,SRRC} = V_{LOS,SRRC} - V_{LOS,aircraft}, \qquad (13)$$

### 5.1 Transmission characteristics of FPIs from different campaigns

A least-squares nonlinear procedure is applied to each sampled transmission function obtained from the BRAINS field campaign in 2009 and NAWDEX airborne campaign in 2016. Figure 5 illustrates the fits of the transmission functions where the intensities are normalized to the maximum of filter A. The black curves are derived from ground-based atmospheric path (ATMG) measurements during the BRAINS field campaign in 2009. The red and blue curves represent the ground-based (INTG) and airborne internal reference path (INTA) measurements obtained from the NAWDEX campaign in 2016,

respectively. The specific parameters of FPIs are listed in Table 3. The difference between ATMG and INTG is due to the different illumination of the FPI via the atmospheric and internal reference paths. Obviously the FWHM of INTA is broader than that of INTG, which is most likely due to small contamination by atmospheric signal not completely blocked within the A2D optical receiver. Specifically, the atmospheric contamination of the internal reference signal of INTA is caused by the

limited suppression efficiency of the electro-optical modulator incorporated in the A2D front optics. This leads to a leakage of atmospheric backscatter being incident on the Rayleigh accumulated charge coupled device (ACCD), during the acquisition time of the internal reference signal. Please note that the internal path signal is recorded with the same ACCD detector as the atmospheric path signal using an integration time of 4.2 µs. For the internal calibration INTG that was performed on ground, the atmospheric path was blocked manually in front of the receiver which completely avoided atmospheric contamination.

**5.2 Determination of FPIs transmission functions for SRRC**

The most critical part both for ALADIN and for the A2D Rayleigh response calibration is the determination of transmission curves of the FPIs for the internal reference and atmospheric paths, respectively. The modelling of FPIs performance has been discussed in previous studies (McGill et al., 1998; McKay et al., 2000a; McKay et al., 2000b). As for ALADIN, the FPIs transmission curve in the atmospheric path is modelled by a convolution of an Airy function, which describes the transmission

of a perfect FPI, and a tilted top-hat function (Dabas and Huber, 2017). The core idea of this approach using Airy and top-hat function is based on the comparison of predicted and the measured Rayleigh response calibration. The FPIs transmission characteristics cannot represent the actual sensitivity of the Rayleigh receiver at the atmospheric path until the difference of predicted and measured responses coincide within a threshold limit.

Different from ALADIN, where only the transmission curve in the internal reference path can be measured during instrument

spectral registration, the A2D FPIs transmission curves both in the internal reference path and in the atmospheric path were measured in previous campaigns. As listed in Table 4, 5 combinations of FPIs transmission functions derived from different campaigns are used to derive different SRRCs. Since there is no simultaneous dropsonde measurement to provide atmospheric temperature and pressure information for modelling the atmospheric molecular backscattered spectrum during the 3[rd] calibration, the radiosonde dataset at a distance of about 229 km to the calibration region (available at:

http://weather.uwyo.edu/upperair/sounding.html) is used. The sensitivity $\beta_x$ and intercept $\alpha_x$ from fitting SRRCs can give a qualitative comparison with the A2D MRRC. According to Eq. (11), the partial derivative of $\alpha_x$ and $\beta_x$ can be obtained as follows:

$$\frac{\partial \Delta f}{\partial \alpha_{ATM}} = -\frac{1}{\beta_{ATM}}, \tag{14}$$

$$\frac{\partial \Delta f}{\partial \alpha_{INT}} = \frac{1}{\beta_{INT}}, \tag{15}$$

$$\frac{\partial \Delta f}{\partial \beta_{ATM}} = \frac{\alpha_{ATM} - M}{\beta_{ATM}^2}, M \equiv R_{ATM} - \gamma_{ATM} \,, \tag{16}$$

$$\frac{\partial \Delta f}{\partial \beta_{INT}} = \frac{N - \alpha_{INT}}{\beta_{INT}^2}, N \equiv R_{INT} - \gamma_{INT} \,, \tag{17}$$

Using the typical values in the previous studies (Lux et al., 2018), that is, $\beta_{ATM} = 5.8 \times 10^{-4}\,\text{MHz}^{-1}$, $\beta_{INT} = 4.5 \times 10^{-4}\,\text{MHz}^{-1}$, $\alpha_{ATM} = -0.06$, $\alpha_{INT} = -0.001$, and assuming realistic values of $\Delta \beta_{ATM} = 10^{-5}\,\text{MHz}^{-1}$, $\Delta \beta_{INT} = 10^{-5}\,\text{MHz}^{-1}$, $\Delta \alpha_{ATM} = 0.01$ and

$\Delta \alpha_{INT} = 0.01$, it can be seen that the change of intercept $\Delta \alpha_{ATM}$ or $\Delta \alpha_{INT}$ results in frequency differences of about -17 MHz or 22 MHz, equivalent to velocity differences of -2.99 m s$^{-1}$ or 3.91 m s$^{-1}$, respectively. The effect of sensitivity $\Delta \beta_{ATM}$ or $\Delta \beta_{INT}$ on velocity is related to the value of $M$ or $N$. In the case of $M = 0$ or $N = 0$, the change of sensitivity $\Delta \beta_{ATM}$ or $\Delta \beta_{INT}$ result in frequency difference of about -1.8 MHz or 0.05 MHz, equivalent to velocity differences of -0.31 m s$^{-1}$ or 0.009 m s$^{-1}$. Therefore, the retrieval of LOS wind velocity is more susceptible to intercept than sensitivity. The measured responses and

simulated SRRCs including fits of internal reference (red) and the 8$^{th}$ atmospheric altitude bin (blue dashed line, the corresponding height is around 5.7 km) are chosen as example and shown in Fig. 6. Comparing the intercepts of measured and simulated ATM response curves, the 1$^{st}$ and 3$^{rd}$ combination shown in Figs. 6 (a) (c) are underestimated (-0.068, -0.102, respectively), while the 2$^{nd}$ and 4$^{th}$ combination shown in Figs. 6 (b) (d) are overestimated (-0.040, -0.042, respectively). Only the 5$^{th}$ combination, shown in Fig. 6 (e) where the FPI parameters obtained from INTA and ATMG are used for internal

reference and atmospheric response determination, shows similar intercept values (-0.055).

In order to further determine which combination matches best to the actual measured Rayleigh calibration response, the procedure adopted from ALADIN (Dabas and Huber, 2017) is used. Herein, $\varepsilon_R$ is defined as the difference between response from the respective SRRCs and the MRRC. Then, the linear fit of $\varepsilon_R$ as function of $f'$ is made, returning a slope $\varepsilon_{R\_slope}$ and intercept $\varepsilon_{R\_intercept}$ based on Eqs. (18A) – (18B) in (Dabas and Huber, 2017). Ideally, if the results from the SRRC and MRRC

match, $\varepsilon_R$ should be randomly fluctuations about 0 with zero $\varepsilon_{R\_intercept}$ and $\varepsilon_{R\_slope}$. Table 4 also lists the fitting results using 5 different combinations, and it is shown that the 5$^{th}$ combination has second smallest absolute $\varepsilon_{R\_slope}$ and $\varepsilon_{R\_intercept}$, offering the overall consistence with the measured case. Therefore, the 5$^{th}$ combination will be used for initial SRRC determination.

### 5.3 Optimization of FPIs transmission characterization

The comparison of sensitivity and intercept of response calibration, as well as the LOS wind velocity derived from SRRC and

A2D measurements, can intuitively assess the feasibility of SRRC on A2D Rayleigh wind retrieval. Figure 7 (a) (b) shows the comparison of $\beta_{ATM}$ and $\Delta \alpha_{ATM} = \alpha_{ATM} + m_{0,ATM}$ between results from SRRC and A2D Rayleigh channel measurement at 08:33:06 UTC on 23 September 2016, respectively. The LOS wind velocity results from SRRC, MRRC, simultaneous

dropsonde measurements and CDL measurements are presented in Fig. 7 (c). It can be seen that $\beta_{ATM}$ and $\Delta\alpha_{ATM}$ derived from SRRC have similar altitude dependence as the one derived from MRRC, indicating that the atmospheric temperature and pressure effect on the response calibration is described correctly using the SRRC. However, the discrepancy of $\Delta\alpha_{ATM}$ between results from SRRC and MRRC shown in Fig. 7 (b) is obvious, resulting in large discrepancy on LOS wind velocity between SRRC and A2D Rayleigh channel datasets shown in Fig. 7 (c). Taking data from a dropsonde which was released from HALO aircraft at the same location as reference, the LOS results from SRRC is underestimated at a height of 1 km – 8 km where it can be regarded as "clear" Rayleigh wind without Mie contamination, assuming that no aerosols are present in this altitude. Thus, a further optimization of FPIs parameters needs to be implemented as the stability of the optical alignment of the instrument can remarkably influence the performance of the A2D (Reitebuch et al., 2009; Lemmerz et al., 2017; Lux et al., 2018).

Considering the optical path of the A2D Rayleigh channel, the FPI centre frequency is sensitive to the incidence angle of the light. It is a reasonable way to optimize FPI transmission function by fine adjusting the centre frequency of filter A or B for the atmospheric path. The Rayleigh spectrometer is composed of two FPIs which are sequentially coupled. Thus, the reflection of the directly illuminated first FPI is directed to the second FPI. Any incidence angle change before the Rayleigh spectrometer will act similarly on both FPIs. Considering that the initial condition was perpendicular incidence, both FPIs are affected similarly regarding a shift in the centre frequency. Furthermore, as angular shifts of only a few µrad are expected to occur, large angles do not have to be considered. Therefore, it is justified to consider the same offset for both centre frequencies induced by small incidence angle changes. Assuming the centre frequencies of filter A and B have the same offset $\Delta f_0$ compared to the values obtained from ATMG, that is, $\Delta f_0 = \Delta f_{0,A} = \Delta f_{0,B}$, and the FPIs parameters at the internal reference path are regarded as ideal, Figs. 8 (a) and 8 (b) present the effect of $\Delta f_0$ on the sensitivity and intercept of fitting SRRC at each altitude bin, respectively. A cost function $F(\Delta f_0)$ is defined to determine the optimized centre frequency as follows:

$$F(\Delta f_0) = \sum_{i=1}^{N} |V_{LOS,SRRC}(i) - V_{LOS,reference}(i)|, \tag{18}$$

where $V_{LOS,SRRC}(i)$ is the LOS wind velocity derived from SRRC with centre frequency offset of $\Delta f_0$ at altitude bin $i$, $V_{LOS,reference}(i)$ is the LOS wind velocity from simultaneous dropsonde datasets interpolated to the height of A2D Rayleigh channel altitude bin $i$. Herein all available altitude bins of SRRC from $i=1$ to $i=N$ ($N=17$) are used to calculate the cost function $F(\Delta f_0)$ for different $\Delta f_0$. It is noted that altitude bins affected by aerosol or cloud layer are hard to be flagged, unless there are auxiliary information such as CDL measurement. Therefore, these bins affected by Mie contamination are also taken into consideration in the calculation of $F(\Delta f_0)$ calculation.

It can be seen from Fig. 8 (c) that $F(\Delta f_0)$ has its minimum when the centre frequencies of both filter A and B for the atmospheric path increase by 20 MHz, corresponding to the optimization case for LOS wind velocity retrieval using SRRC. The profiles for $\beta_{ATM}$ and $\Delta\alpha_{ATM}$ derived from SRRC with FPIs optimization are shown in Figs. 9 (a) and 9 (b), respectively. Compared to Figs. 8 (a) and 8 (b), the increase of centre frequency of filter A and B ($\Delta f_0 > 0$) results in decrease of $\beta_{ATM}$ and $\Delta\alpha_{ATM}$. As shown in Fig. 9 (c), the LOS wind velocity derived from SRRC with optimized FPIs parameters now fits better to the dropsonde results except for heights below 1 km and at around 9 km where Mie contamination may negatively influence the results.

The derived frequency shift of 20 MHz can basically depend on the alignment of the atmospheric optical path. From the experience from the last 10 years it is known that this alignment is not randomly varying from flight to flight, but changes from campaign to campaign. As the telescope and optical receiver is coupled via free optical path (and not via a fibre), the mechanical integration of the A2D into the aircraft prior to each campaign leads to small variation in position and incidence angle on the spectrometers for each deployment. Thus, a valid response calibration can be used for the entire campaigns period. This is true for both, measured or rather simulated response calibrations. In order to monitor the atmospheric path alignment, the position of the spots generated on the ACCD detector behind each FPI is analysed and serves as information on the alignment during the flight itself and among the flights during the campaigns period. It should be noted that the applied frequency shift is only 20 MHz, which is even less than the frequency separation of successive measurement points during a response calibration (25 MHz) and which corresponds to $1.8\times10^{-3}$ of the FSR of the FPIs.

## 6 Statistical comparison and assessment

A statistical comparison of LOS wind velocities derived from SRRC with other instrument measurements is required to assess the feasibility and robustness of SRRC under various atmospheric conditions. Firstly, the quality control based on an SNR mask derived from the A2D Mie channel is applied (Marksteiner, 2013) to identify invalid winds retrieved from the Rayleigh channel, which retains a significant amount of valid Rayleigh winds via a cloud and ground mask (Lux et al., 2018). Then, based on the matched dates listed in Table 2, the comparisons of LOS wind velocity from dropsonde measurements, A2D Rayleigh channel measurements, and results derived from SRRC with and without FPI optimization are illustrated in Fig. 10. A linear fit to the data points is presented to provide the slope and intercept. The correlation coefficient $r$, bias and standard deviation are also calculated and listed in Table 5. Fig. 10 (a) illustrates the comparison of LOS wind velocity between dropsonde and A2D Rayleigh channel measurement, showing that the fit parameters slightly deviate from the ideal case. The correlation coefficient $r$, bias and standard deviation of the A2D Rayleigh winds are 0.95, 0.23 m s$^{-1}$ and 2.20 m s$^{-1}$, respectively, which is comparable to results in previous studies (Lux et al., 2018). The comparison of LOS wind velocity between dropsonde measurements and the results derived from SRRC without FPIs optimization is illustrated in Fig. 10 (b). The corresponding

correlation coefficient *r*, bias and standard deviation are determined to be 0.93, -3.32 m s$^{-1}$ and 2.61 m s$^{-1}$, respectively. It can be seen that underestimation of the LOS wind velocity from SRRC without the FPIs optimization is significant, demonstrating the necessity of the FPIs optimization before wind retrieval when using SRRC procedure. Figure 10 (c) shows the comparison of LOS wind velocity between dropsonde measurements and results derived from SRRC with FPIs optimization. The bias is

0.05 m s$^{-1}$, which is better than the results from A2D wind with MRRC, and the correlation coefficient *r* and standard deviation are 0.94 and 2.52 m s$^{-1}$, respectively. This is comparable to the results from A2D Rayleigh channel measurements and implies the feasibility and robustness of SRRC with FPIs optimization on A2D Rayleigh wind retrieval. From now on, only SRRC results with optimized FPI parameters will be discussed.

In order to evaluate the atmospheric temperature effect on response calibration procedure and wind retrieval, Figure 11 (a)

shows the atmospheric temperature difference between SRRC and MRRC firstly, where the red square and blue bar represent the mean bias and standard deviation at each height. The difference of sensitivity and intercept of response calibration between SRRC and MRRC are illustrated in Figs. 11 (b) (c). It can be seen from Fig. 11 (a) that larger discrepancies of atmospheric temperature can be found at about 7 km to 8 km with mean differences of less than 5 K. But for the corresponding differences of sensitivity and intercept shown in Figs. 11 (b) (c), larger discrepancies appear in lower heights, especially at heights lower

than 3 km. On the one hand, it is implied that the atmospheric temperature effect is less significant in the statistical analysis of 2016 flight campaign. On the other hand, due to the ground elevation during A2D instrument response calibration, the measured response calibration below 2 km in this case cannot be obtained, thus the measured response calibration at height of 2 km are used for LOS velocity retrieval below 2 km, causing larger discrepancies shown in Figs. 11 (b) (c).

The height-dependent comparisons of LOS wind velocity from different datasets after quality control are illustrated in Fig.

12. The mean difference of LOS wind velocity between SRRC and A2D Rayleigh channel measurements shown in Fig. 12 (a) has opposite trend at lower and higher heights, which is related to the intercept difference shown in Fig. 9 (b). Similar LOS wind velocity difference tendency can be seen in Figs. 12 (b) (c) for the case between SRRC and dropsonde, and between A2D Rayleigh channel measurement and dropsonde, respectively. The error bars of LOS velocity derived from MRRC and SRRC can be also seen in Figs. 12 (b) and 12 (c), respectively. Generally, larger discrepancies occur at heights of smaller than 2 km

and larger than 8 km. The LOS wind velocities derived from A2D Rayleigh channel measurements have more obvious discrepancies at heights smaller than 2 km compared to the results derived from SRRC. This is consistent with the fact that inappropriate values of A2D calibration parameters at lower height result in additional LOS velocity bias, and this is one of the limitations of the A2D MRRC approach which can be overcome using the SRRC approach. In order to analyse the height-dependent deviations more comprehensively, Fig. 13 shows the examples of LOS wind velocity from A2D Rayleigh channel

measurement, dropsonde measurements, SRRC and CDL on 23 September 2016, where dropsonde and CDL are interpolated to the A2D height. The CDL provides high performance with accuracy of <0.3 m/s and precision of <1 m/s, respectively

(Chouza, F. et al., 2016), thus we prefer to plot no error bars to the CDL measurements. Larger discrepancies can be obviously seen at heights larger than 8 km due to the occurrence of cloud layer in these cases.

All matched CDL observations listed in Table 2 are used to assess the probability of Mie contamination on Rayleigh wind results. Figure 14 (a) shows the CDL measurement behaviour where valid (or invalid) signal is represented as 1 (or 0). The Mie contamination fraction $F_{Mie}$, shown in Fig. 14 (b), is defined as the ratio of the number of valid signals to all CDL observation number $N$ (here $N=12$) at each height. Obviously, $F_{Mie}$ at heights of smaller than 2 km and between 7 km and 11 km has high values giving the important cause for the larger discrepancies observed in Fig. 12 and 13. It is also implied that even though quality control mentioned above is used, the applied SNR threshold approach cannot guarantee the accurate removal of Rayleigh wind affected by Mie contamination.

## 7 Summary and conclusion

As the first airborne direct detection wind lidar, the A2D has been deployed in several ground and airborne campaigns over the last 12 years for validating the measurement principle of Aeolus and further improving the algorithm and measurement strategy. The A2D instrument calibration is used to obtain the response calibration function indicating the relationship between the measured signal intensities and the Doppler frequency shift which is proportional to the wind speed. However, the atmospheric and instrumental variability currently limit the reliability and repeatability of the A2D instrument response calibration. For instance, there are some factors affecting the accuracy of response calibration directly during instrument response calibration such as Mie contamination, non-zero vertical velocity, and unavailable response functions for lower altitudes due to high ground elevation. The SRRC is thus presented in this paper to overcome these limitations of MRRC.

The most critical part of SRRC is the determination of the transmission characteristics of FPIs for the internal reference and atmospheric paths, respectively. Different from the method used for the determination of ALADIN FPIs transmission curve in the atmospheric path where a tilted top-hat function is used, the A2D candidate SRRCs using different combinations of FPIs transmission characteristics obtained from different campaigns were calculated and compared to the MRRC firstly. It is found that the combination of FPI parameters obtained from airborne internal reference path measurement and the ground-based atmospheric path measurement are the best to be used for the internal reference and atmospheric response determination by SRRC. Since the stability of the optical properties of the FPIs and the optical alignment of the instrument can remarkably influence the performance of the A2D, a fine tuning of FPIs centre frequency for atmospheric path is performed to optimize the SRRC parameters. It is concluded that when the centre frequencies of both filter A and B for the atmospheric path increase by 20 MHz, the LOS wind velocity derived from SRRC provides the best consistency with the simultaneous dropsonde measurements. The dropsonde profile of the wind velocity was used as reference in this study to obtain an optimized SRRC. However, it would also be possible to use other references such as the ECMWF model or 2 $\mu m$ CDL measurements.

What's more, dropsonde data was used as a reference for statistical comparison of LOS wind velocity since it has the generally best spatiotemporal matching and coverage with the results derived from SRRC. Firstly, the biases of LOS wind velocity derived from SRRC without and with FPIs optimization are -3.32 m s$^{-1}$ and 0.05 m s$^{-1}$, respectively, showing the necessity of FPIs optimization for SRRC wind retrieval. The LOS wind velocity from SRRC with FPIs optimization also provides a standard deviation of 2.52 m s$^{-1}$, showing better accuracy and comparable precision with respect to the results obtained from a conventional (measured) Rayleigh response calibration which yield a bias of 0.23 m s$^{-1}$ and standard deviation of 2.20 m s$^{-1}$. This demonstrates the feasibility and robustness of SRRC on A2D Rayleigh wind retrieval. Furthermore, the height-dependent statistical comparison shows that the biases caused by inappropriate calibration parameters below 2 km due to the limiting ground elevation during A2D instrument response calibrations can be overcome by using SRRC, where the response values over the whole altitude range from the aircraft down to mean sea level can be simulated. The larger biases at heights below 2 km and above 8 km are related to residual Mie contamination on the Rayleigh channel. It is also shown that even though quality control based on SNR is used, the accurate removal of points affected by Mie contamination cannot be guaranteed. This shows the necessity of combination of Mie and Rayleigh channel wind analysis.

It should be noted that the A2D SRRC procedure mentioned in this paper is not a pure "copy" from what is done for ALADIN. There are some significant differences, especially in the generation and update of the transmission characteristics of the FPIs of the Rayleigh receiver for the atmospheric channel. Firstly, as opposed to ALADIN, where only the transmission curve in the internal reference path can be measured during instrument spectral registration, the A2D FPI transmission curves both in the internal reference path and in the atmospheric path were measured in previous campaigns, demonstrating slight deviations between both transmission paths due to the aforementioned reasons. Therefore, different combinations of FPI transmission functions derived from different campaigns can be used to derive different candidate SRRCs. After the comparison of candidate SRRCs with simultaneous MRRC, the most satisfactory combination is used for initial SRRC determination. Secondly, as for ALADIN, the core idea of the updated spectral registration using the Airy and top-hat function is based on the comparison of the predicted one and a MRRC. The FPIs transmission characteristics cannot represent the actual sensitivity of the Rayleigh receiver at the atmospheric path until the difference of predicted and the measured responses coincide within a threshold limit. But for A2D, the optical path characteristic of the A2D Rayleigh channel is considered carefully. The optimization of FPIs transmission characteristics was made by fine tuning the centre frequency of filter A or B for the atmospheric path, thus obtaining optimized SRRC.

Overall, the SRRC allows correction for variability in atmospheric temperature and pressure profiles, giving accurate wind retrieval especially in cases of large atmospheric temperature differences between the acquisition time and location of the MRRC and the actual wind measurements. It can also overcome the possible ground elevation limitations, improving the accuracy of A2D wind measurements at lower altitudes. Therefore, it can improve the reliability and repeatability caused by atmospheric and instrumental variability during A2D MRRC process. Further studies based on A2D SRRC will be performed

regarding the atmospheric temperature/pressure effect, Mie contamination correction and the particulate optical properties retrieval.

*Data availability.* Data used in this paper can be provided upon request by email to Oliver Reitebuch (oliver.reitebuch@dlr.de).

*Author contribution.* Xiaochun Zhai prepared the manuscript, developed the method and performed the analysis of the A2D data. Oliver Reitebuch supported the development of the method and analysis of the data. Benjamin Witschas provided the 2-μm DWL data and provided the A2D FPI parameters. Fabian Weiler analyzed the dropsonde data and supported the A2D analysis. Uwe Marksteiner and Oliver Lux provided tools for the analysis of the A2D observations. Benjamin Witschas performed the 2-μm DWL observations, Oliver Lux, Christian Lemmerz and Oliver Reitebuch performed the A2D
measurements. All co-authors provided input to the manuscript and its revision.

*Competing interests.* The authors declare that they have no conflict of interest

*Acknowledgements.* The development of the ALADIN Airborne Demonstrator and the airborne campaigns were supported by the German Aerospace Center (Deutsches Zentrum für Luft- und Raumfahrt e.V., DLR) and the European Space Agency (ESA), providing funds related to the preparation of Aeolus (WindVal II, contract no. 4000114053/15/NL/FF/gp). The first
author was funded by the Chinese Scholarship Council (CSC number: 201706330031).

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

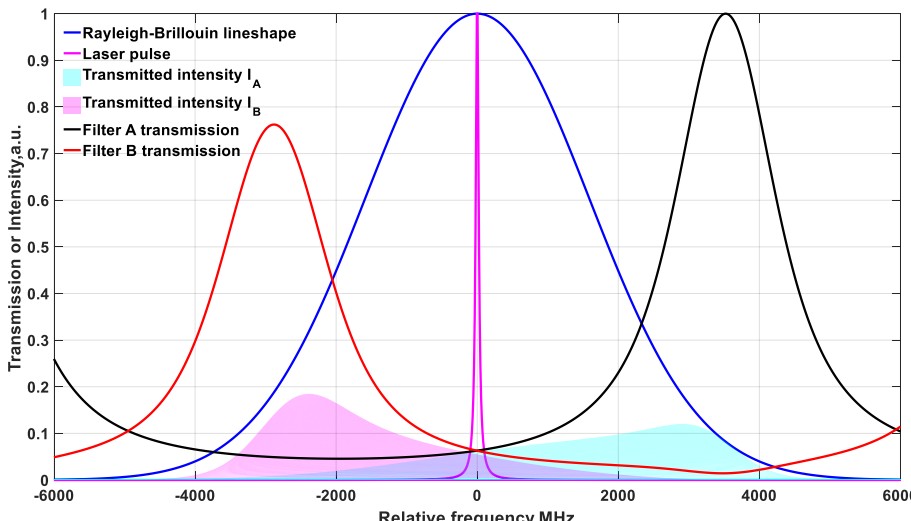

**Figure 1: Modelled spectral distribution of the transmitted laser pulse (pink line) and pure molecular backscatter (blue line) for T=270 K, P=700 hPa normalized to one. The Rayleigh channel transmission spectra of two FPIs are shown in black $T_A$ (*f*) and red $T_B$ (*f*) lines, respectively. The transmitted integrated intensities through FPI A and B are marked with light blue and magenta filled areas.**

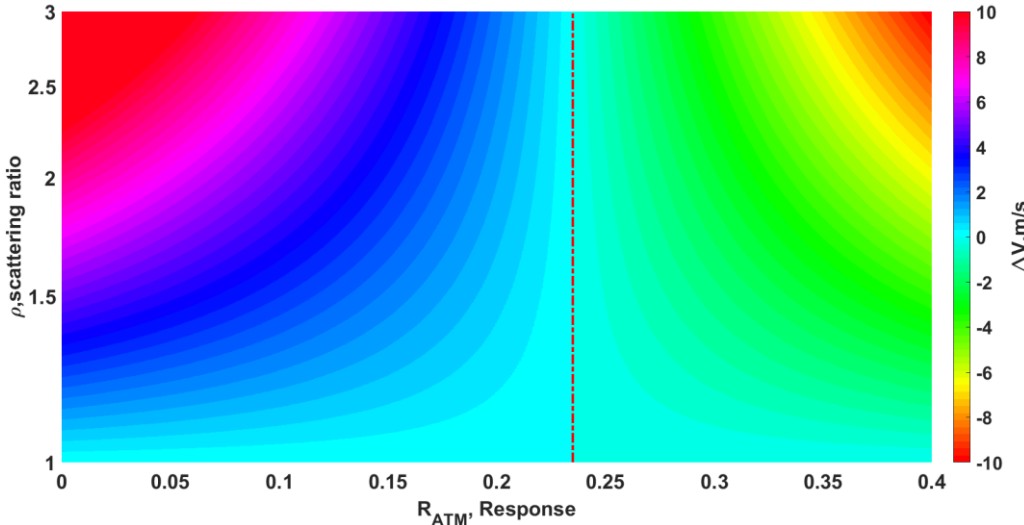

**Figure 2: Simulation of LOS wind velocity errors $\Delta V_{MC}$ induced by Mie contamination and a molecular lineshape at T=223 K and P=301 hPa. The *x*-axis and *y*-axis represent the response value $R_{ATM}$ and scattering ratio $\rho$ , respectively. The red dashed-line corresponds to the response value with minimum $\Delta V_{MC}$ at each scattering ratio.**

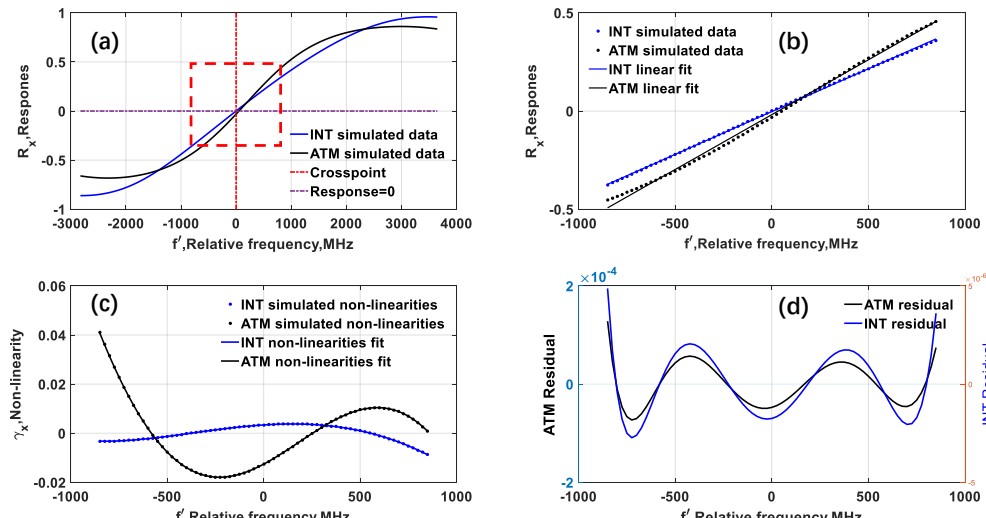

**Figure 3: (a) Simulated Rayleigh Response Calibration (SRRC) for internal reference (INT, blue line) and atmospheric return (ATM, black line), the cross point frequency is marked by red dotted line, (b) INT (blue dots) and ATM (black dots) response functions and corresponding linear least squares fits (blue line for INT, black line for ATM) over a frequency interval of ±850 MHz, where relative frequency is used instead of absolute frequency, (c) simulated non-linearities (dots) and 5th order polynomial fits for INT (blue line) and ATM (black line). (d) response function residuals from INT (blue line) and ATM (black line).**

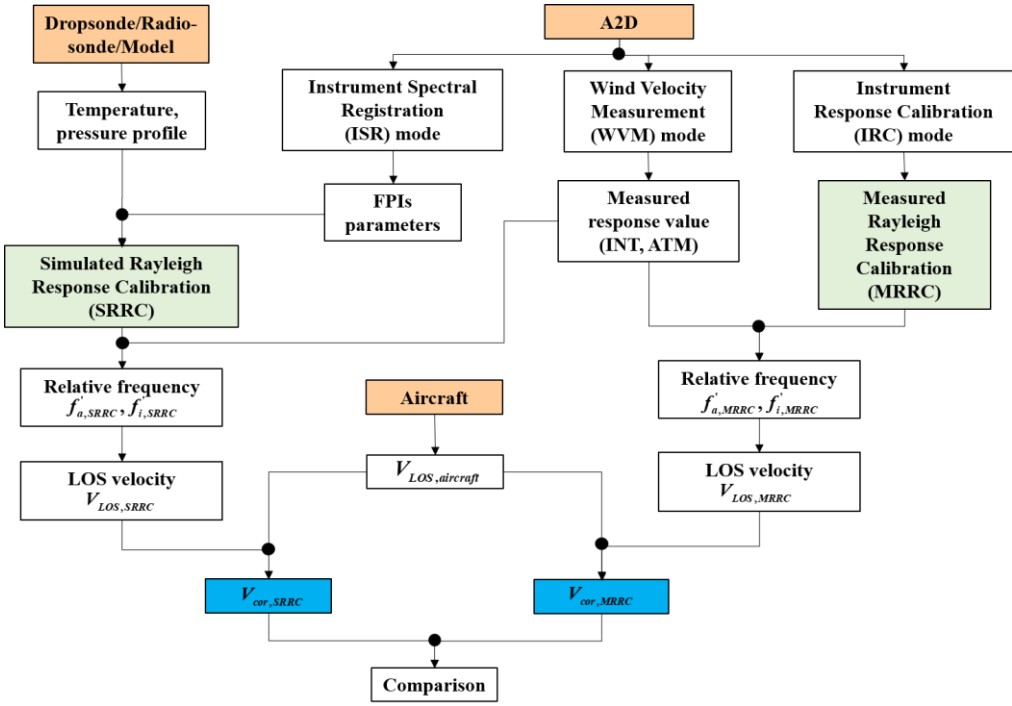

**Figure 4: Flowchart of LOS velocity retrieval and comparison between A2D SRRC and MRRC.**

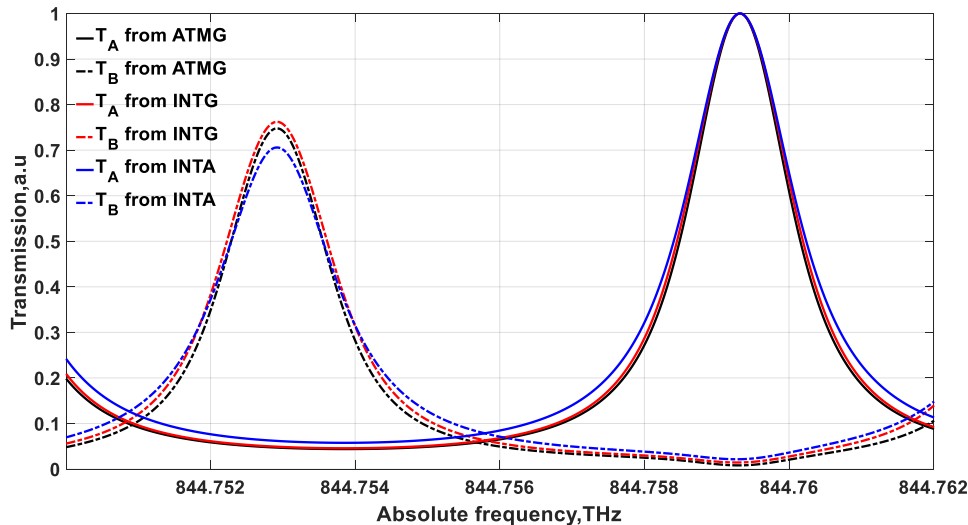

**Figure 5: The transmission function of fitted FPIs from different campaigns and detection channels. The black, red and blue groups are obtained from ATM path measurement during BRAINS ground campaign (ATMG) in 2009, INT path measurement during NAWDEX from ground (INTG) in 2016 and INT path measurement during NAWDEX airborne measurement (INTA) in 2016, respectively.**

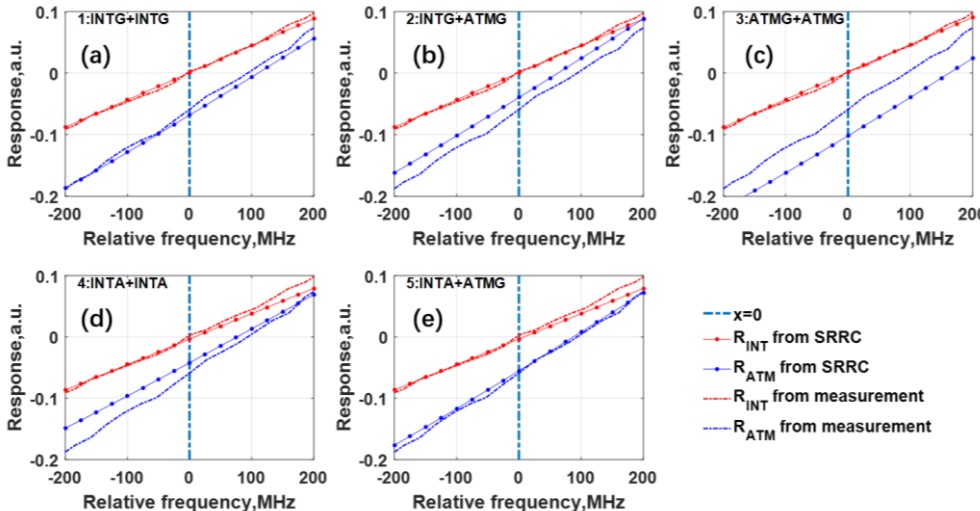

**Figure 6: The response functions of internal reference and 8th atmospheric altitude bin from MRRC (red and blue dashed-lines, respectively, same on every plot) and different SRRCs using different combinations of FPIs transmission parameters (red and blue dotted lines, respectively) a listed in Table 4.**

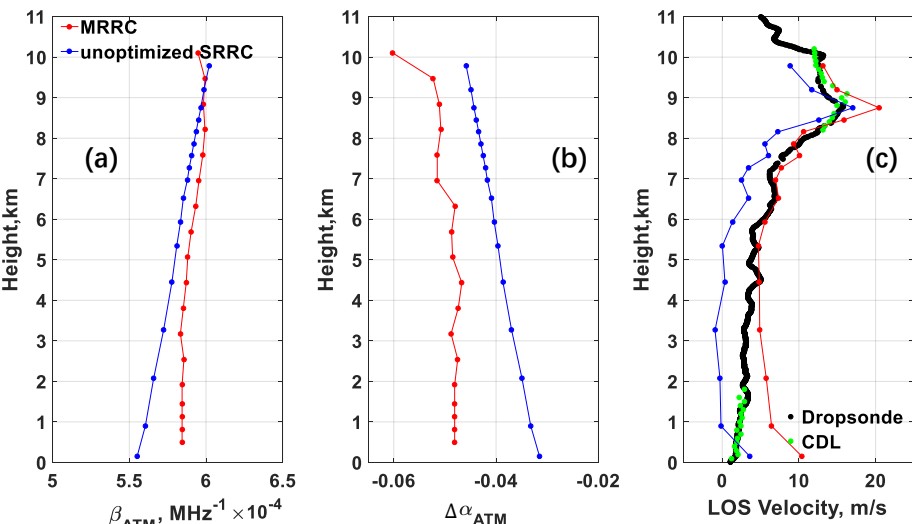

**Figure 7: Case study using dropsonde data on 08:27:07 UTC, 23 September 2016: Comparison of (a) sensitivity $\beta_{ATM}$ (MHz$^{-1}$) (b) $\Delta\alpha_{ATM}$ (c) LOS velocity between results from A2D Rayleigh channel MRRC (red) and not optimized SRRC (blue). The LOS velocity from dropsonde (black) and CDL (green) are also presented in Fig. 7 (c).**

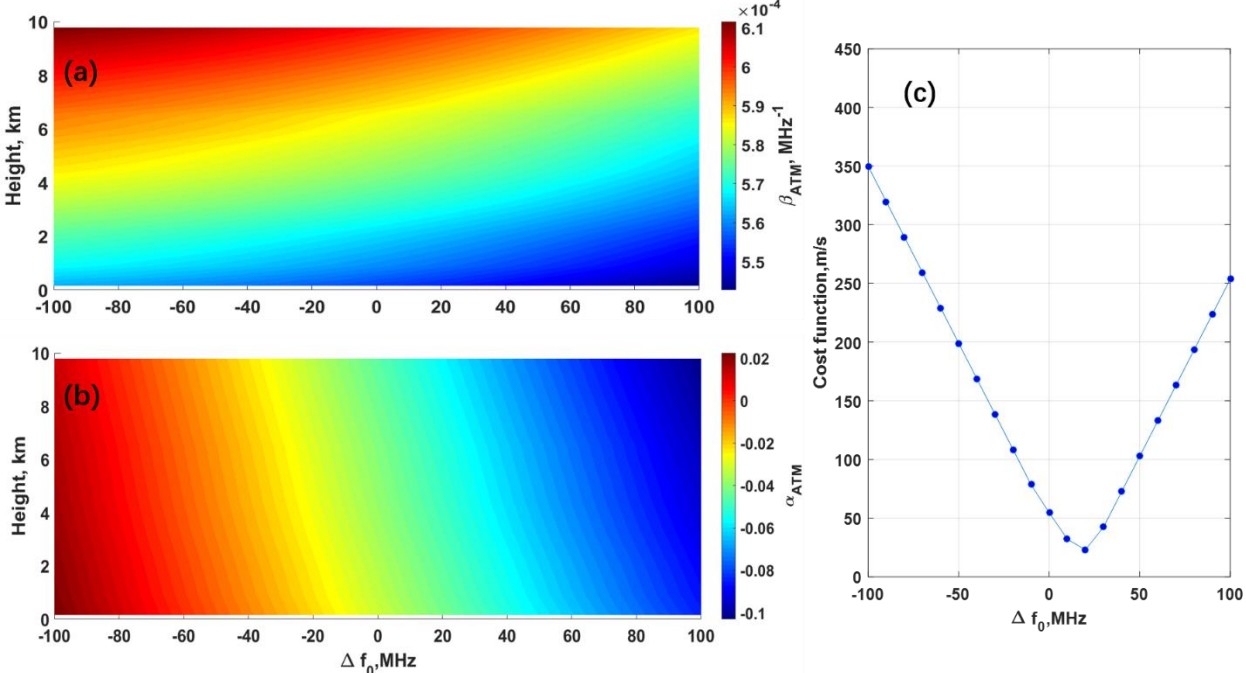

**Figure 8: The effect of the centre frequency offset $\Delta f_0$ of filter A and B for atmospheric path on atmospheric response (a) $\beta_{ATM}$ (b) $\alpha_{ATM}$ and (c) corresponding cost function $F(\Delta f_0)$.**

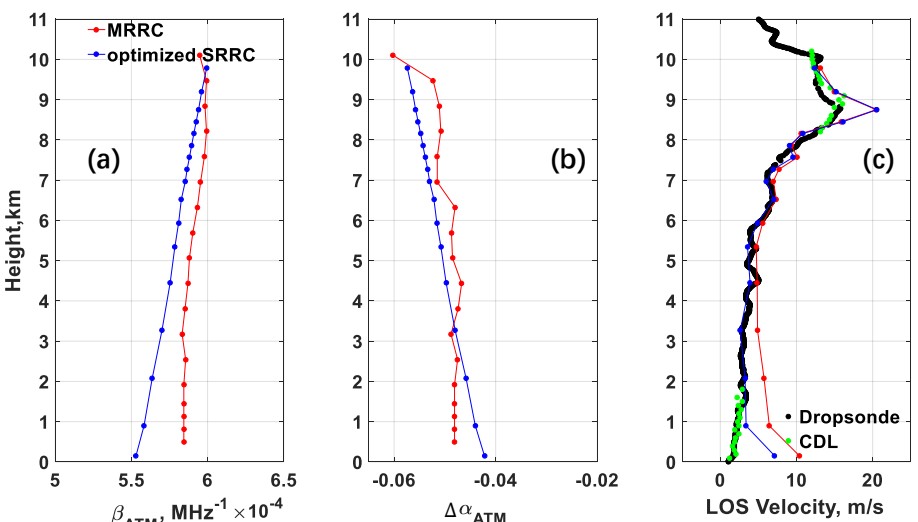

**Figure 9: Case study using dropsonde data on 08:27:07 UTC, 23 September 2016: Comparison of (a) sensitivity $\beta_{ATM}$ (MHz⁻¹) (b) $\triangle\alpha_{ATM}$ (c) retrieved LOS velocity between results from A2D Rayleigh channel MRRC (red) and optimized SRRC (blue). The LOS velocity from dropsonde (black) and CDL (green) are also presented in Fig. 9 (c).**

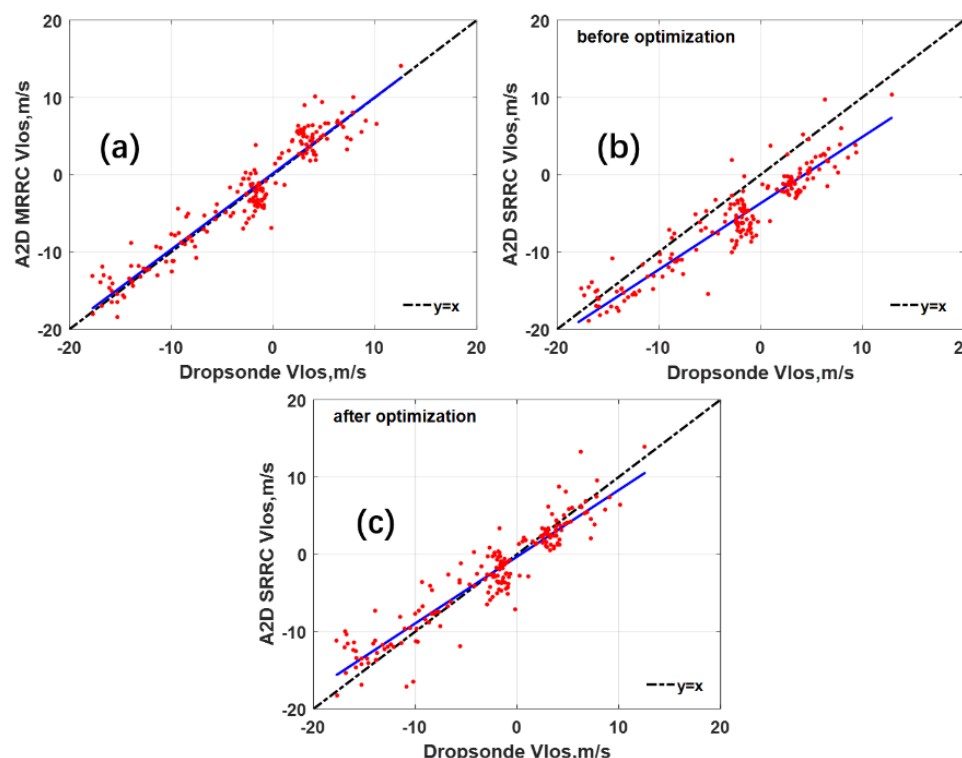

**Figure 10: LOS velocity comparison obtained from (a) dropsonde and A2D Rayleigh channel measurement with MRRC (b) dropsonde and SRRC before FPIs optimization and (c) dropsonde and SRRC after FPIs optimization.**

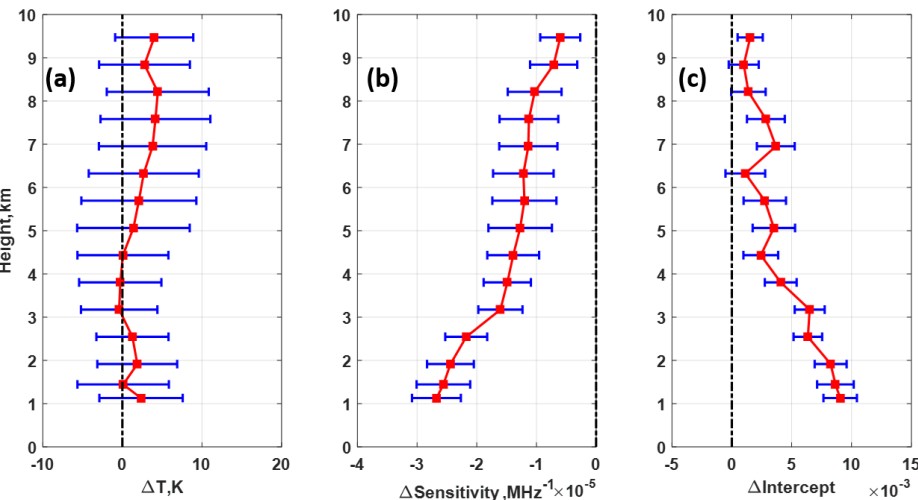

**Figure 11: (a) Difference of temperature between dropsondes used in SRRC and the one during A2D instrument response calibration, and the difference of (b) sensitivity (c) intercept derived from A2D SRRC and MRRC. The red square and the blue bar represent the mean bias and standard deviation at each height.**

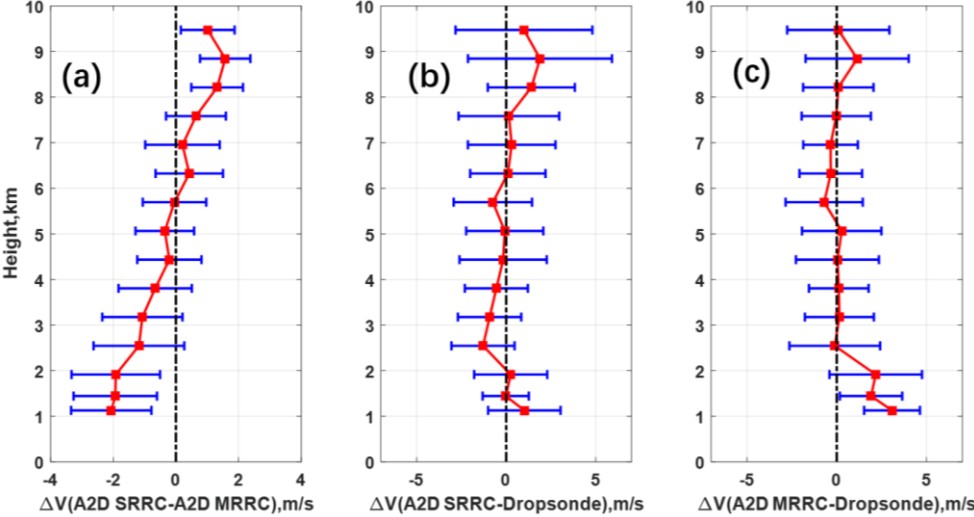

**Figure 12: Comparison of profiles for LOS velocity (a) between A2D SRRC and MRRC (b) SRRC and dropsonde (c) MRRC and dropsonde.**

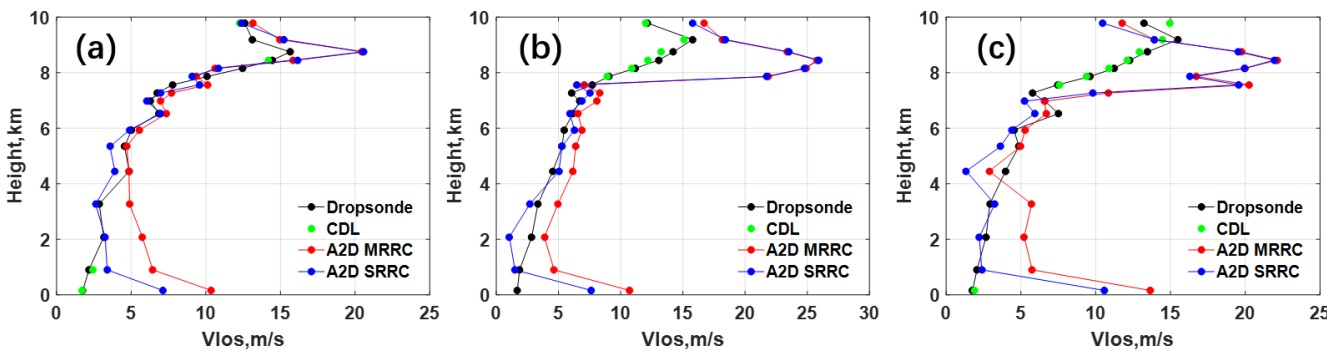

**Figure 13: LOS velocity from dropsonde (black), CDL (green), A2D MRRC (red) and A2D SRRC (blue) on (a) 08:27:07 UTC (b) 08:33:06 UTC (c) 08:39:05 UTC, 23 September 2016.**

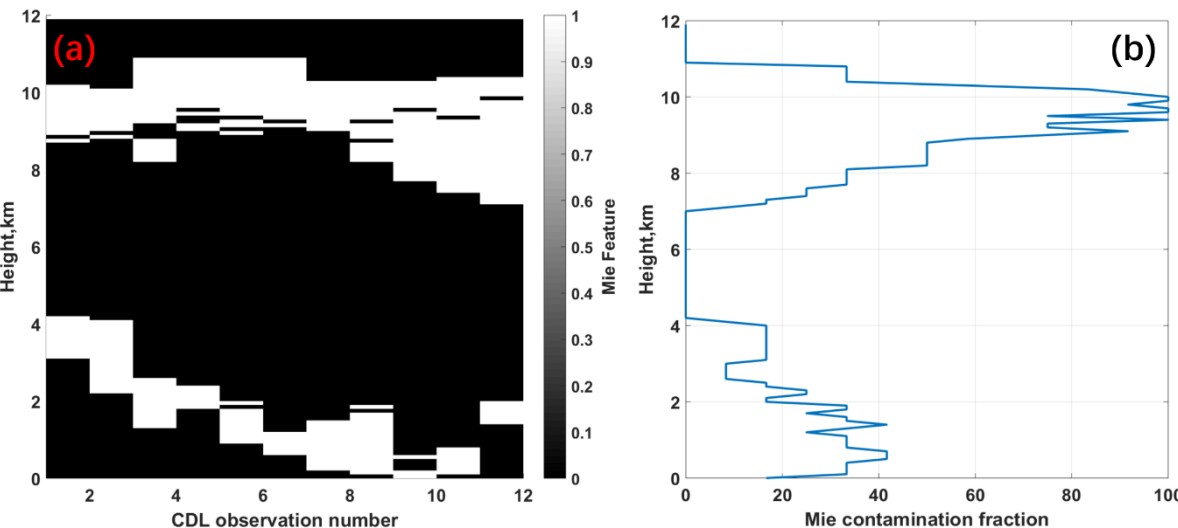

5     **Figure 14: (a) Matched CDL measurement where valid (or invalid) signal is represented as 1 (or 0) (b) Mie contamination fraction $F_{Mie}$ of selected datasets from Table 2 used for comparison.**

**Table 1. Comparison of different FPI-based direct detection wind lidars**

| Lidar | Wavelength and system | Calibration approach | Instrument drift via correction | References |
|---|---|---|---|---|
| OHP [a] Rayleigh lidar | 532 nm, double FPIs | Simulation, FPI scan | quick wind acquisition cycle strategy | Chanin et al., 1989; Garnier and Chanin, 1992; Souprayen et al., 1999a, 1999b |
| NASA [b] Rayleigh/Mie lidar | 355 nm, three FPIs | Simulation, FPI or laser frequency scan | locking etalon and servo-control system | Korb et al., 1992; Korb et al. 1998; Flesia and Korb, 1999; Gentry et al. 2000 |
| USTC [c] Rayleigh lidar | 355 nm, three FPIs | measurement and simulation, FPI scan | locking etalon and servo-control system | Xia et al., 2012; Dou et al., 2014 |
| ESA ALADIN | 355 nm, double FPIs for Rayleigh channel | level 1B: measurement, laser scanning level 2B: simulation, laser frequency scan | internal reference path | Reitebuch et al., 2018; Rennie et al., 2017 |
| DLR A2D | 355 nm, double FPIs for Rayleigh channel | Measurement, laser frequency scan | internal reference path | Marksteiner, 2013; Lux et al., 2018; Marksteiner et al., 2018 |

[a] Observatory of Haute Provence, France

[b] National Aeronautics and Space Administration, U.S.

5    [c] University of Science and Technology of China, China. This lidar is mobile.

**Table 2: Overview of analysed datasets from A2D, 2 μm CDL and dropsonde in the frame of the NAWDEX campaign.**

| Date | A2D measurement period (UTC) and mode | Data availability of CDL | Matched dropsonde Time (UTC) |
|---|---|---|---|
| 17.09.2016 | 10:30-11:35 Wind measurement | available | 11:09:15 11:33:47 |
| | 11:42-12:24 Wind measurement | no data | 11:56:00 12:05:20 12:15:02 12:24:23 |
| 21.09.2016 | 15:34-15:57 Wind measurement | available | 15:40:49 15:45:07 15:48:34 15:52:51 |
| 23.09.2016 | 07:51-08:53 Wind measurement | available | 08:19:01 08:27:07 08:33:06 08:39:05 08:45:05 08:51:16 |
| 28.09.2016 | 12:53 - 13:17 Calibration | available | No data |
| 18.10.2016 | 09:20-09:57 Wind measurement | not available | 09:22:48 09:27:15 09:31:53 09:36:29 09:52:30 |

**Table 3: Specific parameters of FPIs during different ground and airborne campaigns illustrated in Fig. 5**

| Parameters | ATM Ground ATMG | | INT Ground INTG | | INT Airborne INTA | |
|---|---|---|---|---|---|---|
| Filters | filter A | filter B | filter A | filter B | filter A | filter B |
| FSR (GHz) | 10.934 | 10.998 | 10.934 | 10.851 | 10.934 | 10.934 |
| FWHM (GHz) | 1.671 | 1.733 | 1.743 | 1.847 | 1.833 | 1.943 |
| R | 0.670 | 0.696 | 0.668 | 0.679 | 0.622 | 0.610 |
| $\sigma_g$ (MHz) | 266 | 363 | 303 | 391 | 210 | 247 |

5   **Table 4: Combinations for internal reference and atmospheric response simulation with $\varepsilon_{R\_slope}$ and $\varepsilon_{R\_intercept}$ based on Eqs. (18A) – (18B) (Dabas and Huber, 2017).**

| Combination | Internal reference response | Atmospheric response | $\varepsilon_{R\_slope}$ | $\varepsilon_{R\_intercept}$ |
|---|---|---|---|---|
| 1 | INTG | INTG | $-1.48\times10^{-5}$ | -0.0057 |
| 2 | INTG | ATMG | $-1.42\times10^{-7}$ | 0.0206 |
| 3 | ATMG | AMTG | $-1.39\times10^{-5}$ | -0.0356 |
| 4 | INTA | INTA | $-7.74\times10^{-5}$ | 0.0181 |
| 5 | INTA | ATMG | $-9.02\times10^{-7}$ | 0.0059 |

**Table 5: Statistical comparison between results from dropsonde, A2D Rayleigh channel measurement and SRRC before and after FPIs optimization during 2016 campaign.**

| Statistical parameters | Dropsonde to A2D MRRC | Dropsonde to A2D SRRC before FPIs optimization | Dropsonde to A2D SRRC after FPIs optimization |
|---|---|---|---|
| Number of compared data pairs | 185 | 190 | 190 |
| Correlation coefficient, $r$ | 0.95 | 0.93 | 0.94 |
| Slope | 0.99 | 0.86 | 0.86 |
| Intercept, m s$^{-1}$ | 0.19 | -3.70 | -0.32 |
| Mean bias, m s$^{-1}$ | 0.23 | -3.32 | 0.05 |
| Standard deviation, m s$^{-1}$ | 2.20 | 2.61 | 2.52 |