# Peer review of "Rayleigh wind retrieval for the ALADIN airborne demonstrator of the Aeolus mission using simulated response calibration"

_Atmospheric Measurement Techniques, 2019_

## Referee Comment (RC1) · Anonymous Referee #1 · 22 Aug 2019

The paper shows how calibration curves for a direct-detection airborne Doppler lidar can be derived from the known pressure and temperature in the sensed atmospheric volume and a carful characterization of the transmission characteristics of the interferometers used in the receiver. The calibration procedure is copie from what is done for AEOLUS. It is shown that the procedure can be applied as well to the airborne demonstrator of AEOLUS, and achieves a better accuracy with a reduced bias and equivalent standard deviation with colocated drop-sonde wind measurements as with a measured response curve that does not take specifically into account the pressure and temperature conditions.

[Figure]

The practical significance of the method should be discussed. The paper suggests the transmission characteristics of the two FPs are very stable, except for a frequency shift caused by an incidence angle varying from one flight to the other. In Fig 10 or 13, the results are obtained with a frequency shift determined from data acquired during the same flight. Will the frequency shift be significantly modified during another flight? If yes, this should be stressed and a conclusion should be that response calibration should be done every flight.

The SRRC reduced the bias, but on the other hand lower the correlation coefficient with dropsonde vlos in Fig 10. This should be commented.

The paper mentions the presence of an internal reference channel without explaining exactly what it is. A simple graph showing the internal reference and the atmospheric path would improve the clarity of the paper.

Page 2, line 9: in the CDL, the backscattered light captured by the telescope is mixed with a frequency shifted emitter laser. The frequency shift enables the measurement of positive and negative winds. It is not mentioned.

Equations 3 and 4: there integrals should be between $-\infty$ and $+\infty$. In practice $S_a$ has a limited width so the limits $-FSR/2$ $+FSR/2$ can be enough if FSR is much larger, but $+-\infty$ is better.

Page 11, lines 13-20: it is suggested the atmospheric and internal characteristics of FP transmissions are solely due to plate defects. This is wrong. The main reason is the beam étendue is different in the two channels due to a diaphragm.

Page 12, lines 19-23: the authors should write what $\epsilon_R$ is. It is the difference between the SRRC and the MRRC. Ideally it should be randomly fluctuations about 0 with no offset not slope.
* * *

---

## Referee Comment (RC2) · Anonymous Referee #2 · 12 Oct 2019

This paper presents an alternative technique for retrieving LOS wind estimates from the molecular channel of the Aeolus Airborne Demonstrator (A2D) using modeled response functions ("Simulated Rayleigh Response Calibration" or SRRC) derived using best-fit instrument models and the given atmospheric conditions (temperature and pressure) when available from other observations.

The SRRC approach provides some advantages over the "traditional" double-edge approach of measuring calibration response curves during the test process (the MRRC approach), but the authors could do a better job of explaining the reasoning behind this (vs. just listing numbers) at the beginning of the paper and in the abstract. The ap-

proach is a good idea, especially when faced with consistent Mie contamination during flight tests. Have other double-edge wind lidar researchers done anything similar to this before?

An explanation of the physical differences between the internal reference channel and the atmospheric channel would be helpful. For example, does the IRC have a different set of field angles into the FP etalons than provided by the telescope/receive path returns? Does the IRC only see narrowband light?

The paper would also benefit from a short, clear discussion on the topic of Mie (aerosol) contamination on the Rayleigh calibration as the topic comes up several times in the paper. Present the reasons for the aerosol induced bias and reference the literature. This could be followed by cleaning up some paragraphs that vaguely refer the issue, without explaining it.

Some additional proofreading for english language/grammar should catch some minor errors. Remaining comments listed by page/line#.

A "Fair" rating is listed under Scientific Quality because it's not quite at the "Good" level with respect to referencing related work and being clear on the issues addressed, but with minor improvements as listed above and in the following comments, it will likely be above good.

Overall, this is an interesting and useful paper for the field of double-edge direct detection Doppler wind lidar systems.

Page 2 line 24-26: This sentence describing Aeolus is awkward. Perhaps reword as "The novel combination of these two techniques, integrated for the first time into a single wind lidar, expands the observational altitude range from the ground to the lowermost 30 km of the atmosphere." Line 29: Can delete the words, "as well" from the end of the sentence since it begins with "Furthermore".

Page 3 Line 9: Can the authors expand a little bit on the causes of ". . .the atmospheric

and instrumental variability" for readers not familiar with the observation approach. For example, how atmospheric pressure/temperature impact the MRRC and what varies in the instrument (temperature impacting alignment? Variations in the field of view/field angles entering the etalon? Etc.?) Line 12: update to read, "It is based on an accurate theoretical model of the FPI transmission function. . .." Line 28: edit to read, "Table one lists FPI-based direct detection wind lidar systems that are capable of measuring wind information. . .." Note that not all existing FPI systems that can be modeled this way are listed in the table, there are others in existence.

Page 4 Line 10 – Should be "atmospheric conditions"

Page 5 Line 19: fix to read, ". . .the transmission functions of the FPs for the atmospheric path are slightly different compared to . . ." Then please explain why this is (physics causing the differences). Line 24: "regardless of measurement or simulation method, any angular alignment drift will change the incidence angles on FPIs, and hence change their transmission characteristics." Technically, the FPI transmission characteristics should be a function of incidence angles, field of view, temperature, pressure, thickness or gap length, finesse, etc. so perhaps the better term here (and elsewhere) is to say that "any angular alignment drift will change in the incidence angles on the FPIS, resulting in a different transmission value." (or something similar).

Page 6 Line 5: This is an unusual mix of variables (wavelength and frequency shift), but ok. Line 17 and 19: The authors state that Equations 3 and 4 represent convolutions, but this is not mathematically so. These are integrations over frequency of the product of the FPI transfer function times the specific input spectrum value at that frequency. Likewise, integrating this product over only one free spectral range implies that the authors assume there are never any signals outside the etalon FSRs (e.g. where the etalon can start to transmit again). This may be practically true for most applications/wind speeds/platform pointing motions, etc. but should at least be stated as an assumption.

Page 7 Line 5: defects could be in the FPI mirror surface(s) (plural) right? Line 7: Why not also mention/reference the works of Spinhirne, McGill Line 9: R is the mean reflectivity of the etalon mirrors? (again, plural?) Line 14: Suggest instead to say, "An easily calculated analytical expression…." Lines 16-21: The paper might read more easily if this paragraph was moved up earlier in the discussion. Line 21: The "magenta" filled area appears more "pink" – perhaps use that term instead, or "light magenta"

Page 8 Line 1: Here the authors could clarify for the readers not familiar with double-edge approach why the biases are worse when Mie signal is significant but not good enough to measure winds using the Mie channel. Can this be shown somehow in Figure 2? Line 4 (paragraph 2): clarify that the procedure is done assuming no Mie interference (or otherwise?) Line 9: The text says that the red-square marks +/- 850 MHz, but the figure looks like its closer to 1 GHz. Please rectify one or the other to match. Line 22: clarify that the "Then the fit of the SRRC for the internal reference and atmospheric paths can be expressed as a sum of a linear fit plus a 5th order polynomial:"

Page 9 Line 5: replace "In the frame of…" with "As part of.." The rest of this paragraph would benefit from additional proofreading for English grammar. Line 19-20: Suggest a rewrite to read "Time-space matching datasets between dropsonde and A2D can be used as both references to validated A2D wind measurements and to provide essential…." Line 23-24: This sentence repeats a little bit of what was written before, now referring to "illumination properties" - can you be more specific? Is this a function of differences in the spatial (e.g. the pupil) distribution or in the field (e.g. angular) distribution?

Page 10 Line 13-14: The authors state that the, "measured response values obtained from A2D wind velocity measurement mode are brought into the fitted SRRC…." What does "brought into" mean here? Is this a mapping? What is the process for doing this? Line 19-20: Add "and possible vertical velocity components" to the end of this

first sentence.

Page 11 Paragraph 5.1: The figures described here would benefit from a diagram showing the campaign configuration. Line 8-10: This mentions of the difference between ATMG and INTG due to different illumination. The reasoning for this should be described earlier in the paper and referenced back. The authors seem to change terminology back and forth throughout this section (and the corresponding figures) which makes reading the section slightly more challenging. Specifically, on page 8 the terms defined in Equations 8 and 9 are referred to as beta=sensivity and alpha = intercept, but in Figures 7 and 9 only the terms sensitivity and intercept are used. Perhaps adding the variable names beta_ATM and delta-alpha_ATM to the captions for figure 7 and 9 would help. Likewise add the descriptive terms to the caption for figure 8. Line 10: What is the source of the atmospheric signal in the internal path on airborne testing (INTA)? Is there a delay in the internal reference path that causes the INTA signal to overlap with near field returns due to early overlap? Does multiple scattering play a role in these early returns? Lines 13-20: There are numerous papers discussing modeling of FPI performance. Perhaps some of these could also be referenced: • Jack A. McKay and David J. Rees "High-performance Fabry-Perot etalon mount for spaceflight," Optical Engineering 39(1), (1 January 2000). https://doi.org/10.1117/1.602361 • P. D. Atherton, N K. Reay, J. Ring, and T. R. Hicks "Tunable Fabry-Perot Filters," Optical Engineering 20(6), 206806 (1 December 1981). https://doi.org/10.1117/12.7972819 • J.A. McKay and David Rees , "Space-based Doppler wind lidar: Modeling of edge detection and fringe imaging Doppler analyzers" • Others by McKay, and Spinhirne, McGill, Gentry, etc. Line 21: What is meant by the phrase, "Different from ALADIN" ? Were the ALADIN transmission curves (internal and atmospheric paths) never measured?

Page 12 Equations 15 and 17 define variables "A" and "B" for the Atmospheric and Internal paths, but this terminology is confused with the use of those variables as names for "Filter A" and "Filter B" (the two edge filters) per the labeling in Figures 1, 5, etc.

[Figure]

Page 13 Lines 11-13: This information could also be included in a previous section on the impact of angles on FPI transmission functions. Line 13: "Assuming the center frequencies of filter A and B have the same offset..." Are there any challenges to this assumption? If angles get larger, does the center frequency shift more for A vs. B? A diagram (or a reference to a paper with a diagram) of the two paths through the system might help confirm that the offset is the same. Lines 15-20: The text refers to the plots in Figure 8 and talks about range gates, but the figure shows altitude bins. Which terminology should be used? Line 20: "all available range gates . . . are used to calculate the cost function..." – does this assume there is no aerosol present in this data set?

Page 14 Lines 28-29: The sentence, "However, the temperature difference between MRRC and the actual wind measurement must..." is confusing. Perhaps the authors meant, "However, differences in the atmospheric temperature profile between when the MRRC was obtained and when the actual wind measurements were acquired are a known important source of wind bias, which are especially severe in cases of large temperature differences." Lines 21-33 (and line 11 on page 15): This issue is the basis for all the work done in this paper, right? So this should be right up front in the beginning of the paper, to help the reader understand why the work is being done and described.

Page 15 Line 11: This is the key point of the paper, but it is muddled a little due to grammar. Perhaps say "This is one of the limitations of the A2D MRRC approach which can be overcome using the SRRC approach" Line 15-17: Can you be more specific than saying the response calibration is affected directly? Perhaps say that the aerosol spectrum shifts the centroid of the atmospheric/filter transmission product, thereby biasing the wind speed estimates (or something like that)? Line 25: "Indeed, the Mie contamination...." – this is another key point for the paper and justification for doing the SRRC. While a detailed discussion might not be within the scope of the paper, the paper would benefit greatly from some discussion as the topic of Mie contamination

comes up several time.

Page 16 Line 17: remove the "are" from the beginning of the line. Line 28: "overcame" should be "overcome" here.

Page 17 Line 1: The sentence should probably read, "Overall, the SRRC allows correction for variability in atmospheric and temperature profiles, when known,..."

Figures Figure 2: Please also use the variable name (e.g. "Rx") with "Response" (in the caption and the axis labels) Figure 3: Again, refer to the variable fc when discussing the cross point frequency. Figure 5: Should the blue dashed curve be labeled "TB from INTA" (vs. TA) ? Figure 6: The authors could clarify for the reader that the MRRC lines are repeated through out the plots, e.g. say "(red and blue dashed-lines, respectively, same on every plot)" Figure 9: clarify that (c) represents the retrieved LOS velocity Figure 13: Can the authors say anything about the potential presence of vertical velocities and their impact on the comparison? Can the authors provide error bars on the LOS velocity retrievals? Even CDL systems have errors.

---

## Author Comment (AC3) · 27 Nov 2019

**Response to Referee Comment 1 on "***Rayleigh wind retrieval for the ALADIN airborne demonstrator of the Aeolus mission using simulated response calibration***"**

We appreciate the referee's insightful and valuable comments on our manuscript AMT-2019-274, which helped to significantly improve the manuscript. We have revised the manuscript accordingly. A point-by-point response is also attached to this file.

**Comments:**

1. The paper shows how calibration curves for a direct-detection airborne Doppler lidar can be derived from the known pressure and temperature in the sensed atmospheric volume and a careful characterization of the transmission characteristics of the interferometers used in the receiver. The calibration procedure is a copy from what is done for AEOLUS. It is shown that the procedure can be applied as well to the airborne demonstrator of AEOLUS, and achieves a better accuracy with a reduced bias and equivalent standard deviation with collocated drop-sonde wind measurements as with a measured response curve that does not take specifically into account the pressure and temperature conditions.

R: Thanks for your comment. It should be noted that the A2D SRRC procedure mentioned in this paper is not a pure "copy" from what is done for ALADIN. There are some significant differences, especially in the generation and update of the transmission characteristics of the FPIs of the Rayleigh receiver for the atmospheric channel. The specific differences are listed below:

1. The transmission characteristics of FPIs for the atmospheric path are different from the transmission curves registered on the internal reference path during the instrument spectral registration because of the difference of the illumination of the beams in the atmospheric and the internal reference paths due to different divergence and incidence angles on FPIs (Reitebuch et al., 2009). **As for ALADIN**, the FPIs transmission curve in the atmospheric path is modelled by a convolution of an Airy function, which describes the transmission of a perfect FPI, and a defect function which is used to consider deviations from the perfect FPI. For ALADIN, a tilted top-hat function is used (Dabas and Huber, 2017), whereas a Gaussian defect function is used for the A2D. As opposed to ALADIN, where only the transmission curve in the internal reference path can be measured during instrument spectral registration, **the A2D** FPI transmission curves both in the internal reference path and in the atmospheric path were measured in previous campaigns, demonstrating slight deviations between both transmission paths due to the aforementioned reasons. Therefore, different combinations of FPI transmission functions derived from different campaigns can be used to derive different candidate SRRCs. After the comparison of candidate SRRCs with simultaneous MRRC, the most satisfactory combination is used for initial SRRC determination.

2. **As for ALADIN**, the core idea of the updated spectral registration using the Airy and top-hat function is based on the comparison of the predicted one and an MRRC. The FPIs transmission characteristics cannot represent the actual sensitivity of the Rayleigh receiver at the atmospheric path until the difference of predicted and the measured responses coincide within a threshold limit. But for **A2D**, the optical path characteristic of the A2D Rayleigh channel is considered

carefully. Basically, the FPI center frequency is sensitive to the incidence angle of the light. It is a reasonable way to optimize the FPI transmission function by fine adjusting the center frequency of filter A or B for the atmospheric path. The Rayleigh spectrometer is composed of two FPIs which are sequentially coupled. Thus, the reflection of the directly illuminated FPI is directed to the second FPI. Any incidence angle change in front of the Rayleigh spectrometer will act similarly on both FPIs.

The related description has been added in **Sect 7 Page 20 Line 5-23.**

2. The practical significance of the method should be discussed. The paper suggests the transmission characteristics of the two FPs are very stable, except for a frequency shift caused by an incidence angle varying from one flight to the other. In Fig 10 or 13, the results are obtained with a frequency shift determined from data acquired during the same flight. Will the frequency shift be significantly modified during another flight? If yes, this should be stressed and a conclusion should be that response calibration should be done every flight.

R: revised. Thank you for this comment. We will add a clarification to the manuscript.

The derived frequency shift of 20 MHz can basically depend on the alignment of the atmospheric optical path. From the experience from the last 10 years it is known that this alignment is not randomly varying from flight to flight, but changes from campaign to campaign. As the telescope and optical receiver is coupled via free optical path (and not via a fibre), the mechanical integration of the A2D into the aircraft prior to each campaign leads to small variation in position and incidence angle on the spectrometers for each deployment. Thus, a valid response calibration can be used for the entire campaigns period. This is true for both, measured or rather simulated response calibrations. In order to monitor the atmospheric path alignment, the position of the spots generated on the ACCD detector behind each FPI is analyzed and serves as information on the alignment during the flight itself and among the flights during the campaigns period. It should be noted that the applied frequency shift is only 20 MHz, which is even less than the frequency separation of successive measurement points during a response calibration (25 MHz) and which corresponds to $1.8\times10^{-3}$ of the FSR of the FPIs. The related description has been added to **Sect 5.3 Page 16 Line 17-26.**

3. The SRRC reduced the bias, but on the other hand lower the correlation coefficient with dropsonde vlos in Fig 10. This should be commented.

R: revised. The comparison of the correlation coefficient has been added **in Sect 6 Page 17 line 8-21**: "The correlation coefficient, bias and standard deviation are also calculated and listed in Table 5. Fig. 10 (a) illustrates the comparison of the LOS wind velocity between dropsonde and A2D Rayleigh channel measurements, showing that the fit parameters slightly deviate from the ideal case. The correlation coefficient, bias and standard deviation of the A2D Rayleigh winds are 0.95, 0.23 m s$^{-1}$ and 2.20 m s$^{-1}$, respectively, which is comparable to results in previous studies (Lux et al., 2018). The comparison of LOS wind velocity between dropsonde measurements and the results derived from SRRC without FPIs optimization is illustrated in Fig. 10 (b). The corresponding correlation coefficient, bias and standard deviation are determined to be 0.93, -3.32 m s$^{-1}$ and 2.61 m s$^{-1}$, respectively. It can be seen that the underestimation of the LOS wind velocity from SRRC without the FPIs optimization is significant, demonstrating the necessity of the FPIs optimization before wind retrieval using

SRRC procedure. Figure 10 (c) shows the comparison of LOS wind velocity between dropsonde measurements and results derived from SRRC with FPIs optimization. The bias is 0.05 m s$^{-1}$, which is better than the results from A2D wind with MRRC, and the correlation coefficient and standard deviation are 0.94, 2.52 m s$^{-1}$, respectively, comparable to the results from A2D Rayleigh channel measurements, thus implying the feasibility and robustness of SRRC with FPIs optimization on A2D Rayleigh wind retrieval. From now on, only SRRC results with optimized FPI parameters will be discussed."

4. The paper mentions the presence of an internal reference channel without explaining exactly what it is. A simple graph showing the internal reference and the atmospheric path would improve the clarity of the paper.

R: revised. Thanks for your suggestion, we didn't explain it clearly. The specific schematic of ALADIN Airborne Demonstrator (A2D) was shown in Fig. 1 in (Lux et al., 2018), which has been already referenced in the following added in **Sect 2, page 4, line 11-23** see below:

"For each direct detection wind lidar system, the emitted laser frequency should be known to accurately derive the Doppler frequency shift. A zero Doppler shift reference determined by pointing to the zenith direction has been used to correct for the short-term frequency drift in previous studies (Souprayen et al., 1999b; Korb et al., 1992; Dou et al., 2014). But for the A2D, the internal reference path is specially used to measure the emitted laser frequency information. As shown in Fig. 1 in (Lux et al., 2018), a small portion of laser beam radiation is collected by an integrating sphere and coupled into a multi-mode fibre, then injected into the receiver via the front optics. The atmospheric backscattered signal is collected by a Cassegrain telescope and guided via free optical path propagation to the front optics and receiver successively. This path is called the atmospheric path. An electro-optic modulator is used to separate the atmospheric signal from the internal reference signal temporally in order to minimize the contamination of the internal reference signal with atmospheric signals and saturation of the detectors at short ranges (Reitebuch et al., 2009). Because of the different optical illumination of the internal path and atmospheric path resulting in different divergence and incidence angles on the FPIs, the response calibration curves for these two paths are slightly different. Note that ALADIN uses free path propagation rather than a fibre coupling unit for the internal reference path."

The related descriptions of the internal reference path and atmospheric path are also updated:
1. **In Sect 2.1, page 5, line 9-11**, "The ALADIN Rayleigh winds produced by the level 1B processor (Reitebuch et al., 2018) are based on a MRRC while the level 2B processor uses SRRC. Basically, MRRC includes two response calibration curves derived from internal reference path and atmospheric path, respectively."

2. **In Sect 2.2, page 6, line 25-28**, "The A2D SRRC based on this simulation approach promises an improvement in terms of A2D wind speed errors due to the limitations of A2D MRRC. Similar to MRRC, SRRC also includes two response calibration curves derived from internal reference path and atmospheric path, respectively."

5. Page 2, line 9: in the CDL, the backscattered light captured by the telescope is mixed with a frequency shifted emitter laser. The frequency shift enables the measurement of positive and negative winds. It is not mentioned.
   R: revised. Please see **Sect 1, page 2, line 13-14:** "…, light, and the frequency shift introduced by an acoustic-optical modulator enables the measurement of positive and negative winds."

6. Equations 3 and 4: there integrals should be between -\infty and +\infty. In practice S_a has a limited width so the limits -FSR/2 +FSR/2 can be enough if FSR is much larger, but +-\infty is better.
   R: revised. Please see the updated **Equation 3 and 4 in Sect 3, page 7, line 22-24.**

$$I_{A,B,INT}(f_i) = \int_{-\infty}^{+\infty} T_{A,B,INT}(f)S_i(f_i - f)df$$

$$I_{A,B,ATM}(f_a) = \int_{-\infty}^{+\infty} T_{A,B,ATM}(f)S_a(f_a - f)df$$

7. Page 11, lines 13-20: it is suggested the atmospheric and internal characteristics of FP transmissions are solely due to plate defects. This is wrong. The main reason is the beam étendue is different in the two channels due to a diaphragm.
   R: revised. Thanks for your comment, yes, we didn't explain it correctly at this point. It has been revised as "the transmission characteristics of the FPIs for the atmospheric path are different from the transmission curves registered on the internal reference path during the instrument spectral registration because of slightly different illumination of the beams in the respective paths due to different divergence and incidence angles on FPIs (Reitebuch et al., 2009)." Please see **Sect 2.2, page 6, line 20-24**.

8. Page 12, lines 19-23: the authors should write what \eplison_R is. It is the difference between the SRRC and the MRRC. Ideally it should be randomly fluctuations about 0 with no offset not slope.
   R: revised. The definition of $\varepsilon_R$ has been updated in the revised manuscript as "$\varepsilon_R$ is defined as the difference between response from the respective SRRCs and the MRRC. Then, the linear fit of $\varepsilon_R$ as function of $f'$ is made, returning a slope and intercept based on Eqs. (18A) – (18B) in (Dabas and Huber, 2017). Ideally, if the result from the SRRC matches the measured one from MRRC, it should be randomly fluctuations about 0 with zero intercept and slope", please see **Sect 5.2, page 14, line 26-27 to page 15 line 1-2.**

---

## Author Comment (AC4) · 27 Nov 2019

**Response to Referee Comment 2 on** "Rayleigh wind retrieval for the ALADIN airborne demonstrator of the Aeolus mission using simulated response calibration"**

We appreciate the referee's insightful and valuable comments on our manuscript AMT-2019-274. We thank the referee explicitly for careful proofreading, which significantly improve the clarity of the text and the readability for a broader audience. According to the suggestions and questions, the point-by-point response is attached in this file.

**Major comments:**

This paper presents an alternative technique for retrieving LOS wind estimates from the molecular channel of the Aeolus Airborne Demonstrator (A2D) using modeled response functions ("Simulated Rayleigh Response Calibration" or SRRC) derived using best-fit instrument models and the given atmospheric conditions (temperature and pressure) when available from other observations.

 The SRRC approach provides some advantages over the "traditional" double-edge approach of measuring calibration response curves during the test process (the MRRC approach), but the authors could do a better job of explaining the reasoning behind this (vs. just listing numbers) at the beginning of the paper and in the abstract. The approach is a good idea, especially when faced with consistent Mie contamination during flight tests.
 R: revised.

In the abstract, page 1, line 14-19, the reason why SRRC provides advantages over MRRC is added and revised as "..., However, differences in the atmospheric temperature profile between the location and time of the MRRC and the actual wind measurements are important sources of wind bias since the atmospheric temperature has a direct effect on the instrument response calibration. Furthermore, some experimental limitations need to be considered carefully to achieve a reliable MRRC. The atmospheric and instrumental variability thus currently limit the reliability and repeatability of this MRRC. In this paper, a procedure ... to resolve these limitations of the A2D Rayleigh channel MRRC".

In addition, related introductions are also added in Sect.1, Page 3, line 14-23, "Currently, only measured Rayleigh response calibrations (MRRC) are used for the A2D (Marksteiner, 2013; Lux et al., 2018; Marksteiner et al., 2018). However, the atmospheric temperature affects the Rayleigh-Brillouin line shape, and has a direct effect on the instrument response calibration (Dabas et al., 2008). Differences in the atmospheric temperature profile of the time and location when the MRRC was obtained and the actual wind measurements are important sources of wind bias, which are especially severe in case of large temperature differences. This is the reason why it is mandatory to consider the atmospheric temperature in the Aeolus level 2B procedure to retrieve reliable winds (Dabas et al., 2008; Rennie et al., 2017). Furthermore, some experimental limitations, which will be introduced specifically in Sect. 2.1, need to be considered carefully to achieve a reliable MRRC. Overall, the atmospheric and instrumental variability coming along with an MRRC limits the reliability and repeatability of A2D instrument response calibrations."

**Have other double-edge wind lidar researchers done anything similar to this before?**

R: Yes, as shown in Table 1, there are several FPI-based direct detection wind lidar systems that are capable of measuring wind based on a measurement approach or a simulation approach. The black-marked parts use a simulation approach to obtain calibration response curves, which is similar to the SRRC method mentioned in this paper.

| Lidar                                      | Wavelength and system                             | Calibration approach                                                                      | Instrument
drift
correction              | References                                                                                  |
|--------------------------------------------|---------------------------------------------------|-------------------------------------------------------------------------------------------|------------------------------------------------|---------------------------------------------------------------------------------------------|
| OHP a
Rayleigh lidar         | 532 nm, double
FPIs                            | Simulation ,
FPI scanning                                                       | quick wind
acquisition
cycle strategy    | Chanin et al., 1989;
Garnier and Chanin,
1992; Souprayen et
al., 1999a, 1999b      |
| NASA b
Rayleigh/Mie
lidar | 355 nm, three
FPIs                             | Simulation
FPI or laser
scanning                                                    | locking etalon
and servo-
control system | Korb et al., 1992;
Korb et al. 1998;
Flesia and Korb,
1999; Flesia et al.,
2000 |
| USTC c
Rayleigh lidar        | 355 nm, three
FPIs                             | measurement
and simulation,
FPI scanning                                     | locking etalon
and servo-
control system | Xia et al., 2012;
Dou et al., 2014                                                       |
| ESA
ALADIN                              | 355 nm, double
FPIs for
Rayleigh
channel | level 1B:
measurement,
laser scanning
level 2B:
simulation,
laser scanning | internal
reference path                     | Reitebuch et al.,
2018;
Rennie et al., 2017                                           |
| DLR
A2D                                 | 355 nm, double
FPIs for
Rayleigh
channel | Measurement,
laser scanning                                                            | internal
reference path                     | Marksteiner, 2013;
Lux et al.,2018;
Marksteiner et al.,
2018                       |

**Table 1. Comparison of different FPI-based direct detection wind lidars**

a Observatory of Haute Provence, France

b National Aeronautics and Space Administration, U.S.

c University of Science and Technology of China, China. This lidar is mobile.

2. An explanation of the physical differences between the internal reference channel and the atmospheric channel would be helpful. For example, does the IRC have a different set of field angles into the FP etalons than provided by the telescope/receive path returns? Does the IRC only see narrowband light?

R: revised. Thanks for your suggestion, we didn't explain it clearly. Yes, the atmospheric path and internal reference path differ in their field angles. The internal reference signal is coupled into the receiver via an optical fiber whereas the atmospheric signal enters the receiver via free beam bath through a set of different optics. This leads to a slightly different set of field angles on the FPIs for the internal path and the atmospheric path. During the ISR only the internal path, illuminated with spectrally narrow-band light from the laser is recorded, while for the IRC the internal path (with narrow spectral bandwidth from the laser) and the atmospheric path with broad spectral bandwidth molecular returns, but also narrow spectral bandwidth cloud, aerosol and ground returns is recorded. The specific schematic of ALADIN Airborne Demonstrator (A2D) was shown in Fig. 1 in (Lux et al., 2018), which has been already referenced in the following added paragraph in Sect 2, page 4, line 11-23, see below:

"For each direct detection wind lidar system, the emitted laser frequency should be known to accurately derive the Doppler frequency shift. A zero Doppler shift reference determined by pointing to the zenith direction has been used to correct for the short-term frequency drift in previous studies (Souprayen et al., 1999b; Korb et al., 1992; Dou et al., 2014). But for the A2D, the internal reference path is specially used to measure the emitted laser frequency information. As shown in Fig. 1 in (Lux et al., 2018), a small portion of laser beam radiation is collected by an integrating sphere and coupled into a multi-mode fibre, then injected into the receiver via the front optics. The atmospheric backscattered signal is collected by a Cassegrain telescope and guided via free optical path propagation to the front optics and receiver successively. This path is called the atmospheric path. An electro-optic modulator is used to separate the atmospheric signal from the internal reference signal temporally in order to minimize the contamination of the internal reference signal with atmospheric signals and saturation of the detectors at short ranges (Reitebuch et al., 2009). Because of the different optical illumination of the internal path and atmospheric path resulting in different divergence and incidence angles on the FPIs, the response calibration curves for these two paths are slightly different. Note that ALADIN uses free path propagation rather than a fibre coupling unit for the internal reference path."

The related descriptions of the internal reference path and atmospheric path are also updated:

- In Sect 2.1, page 5, line 9-11, "The ALADIN Rayleigh winds produced by the level 1B processor (Reitebuch et al., 2018) are based on a MRRC while the level 2B processor uses SRRC. Basically, MRRC includes two response calibration curves derived from internal reference path and atmospheric path, respectively."
- In Sect 2.2, page 6, line 25-28, "The A2D SRRC based on this simulation approach promises an improvement in terms of A2D wind speed errors due to the limitations of A2D MRRC. Similar to MRRC, SRRC also includes two response calibration curves derived from internal reference path and atmospheric path, respectively."
- 3. The paper would also benefit from a short, clear discussion on the topic of Mie (aerosol) contamination on the Rayleigh calibration as the topic comes up several times in the paper. Present the reasons for the aerosol induced bias and reference the literature. This could be followed by cleaning up some paragraphs that vaguely refer the issue, without explaining it. R: revised. Thanks for your suggestion. About the topic of Mie contamination on Rayleigh calibration, we have updated related paragraphs.

Firstly, in Sect 2.1, page 5, line 23-27, we discuss the reasons for the aerosol induced bias: "..., the spectrally narrowband Mie scattering which is not filtered out by the Fizeau interferometer will enter the FPIs and can be considered as Mie contamination of the Rayleigh

signal. Because of the different spectral widths of the particle and molecular backscatter signal, the sensitivities of the FPIs on them are different. The Mie contamination on Rayleigh channel is one of the sources for systematic errors because it modifies the instrument response calibration curve, which should be avoided to ensure the representativity of pure Rayleigh response."

Then, we analyze the LOS wind velocity error induced by Mie contamination in Sect 3 page 9 line 9-14 based on simulation results: "The LOS wind velocity error induced by Mie contamination  $\Delta V$  is defined as the difference of LOS wind velocity under pure atmospheric molecular condition and atmospheric spectral condition with scattering ratio of  $\rho$ . Figure 2 shows the simulation for  $\Delta V$  at T=223 K and P=301 hPa, where the x-axis and y-axis represent different response values and scattering ratios, respectively. Positive and negative  $\Delta V$  represent the overestimation and underestimation of LOS velocity, respectively. An overestimation of the LOS velocity occurs at response values less than 0.2. Larger scattering ratios result in a larger overestimation. The difference can get up to 20 m s-1 in case of  $\rho > 10$ , if this Mie-crosstalk is not considered.

We also introduce the effect of Mie contamination correction on systematic error optimization in Sect 3 page 9 line 14-19: "According to previous studies (Dabas et al., 2008), it is implied that the Mie contamination correction could improve the quality of the Rayleigh wind in the cases of intermediate  $\rho$ , e.g. below 1.5, as in this case the Mie signal is not high enough to guarantee an accurate Mie wind measurement but rather becomes significant for the Rayleigh channel (Sun et al., 2014; Lux et al., 2018). The value of  $\rho$ , which is needed to correct for the Mie contamination in the Rayleigh channel, is obtained by analyzing the Mie channel signal (Flamant et al., 2017). "

**Reference:**

 Flamant, P., Lever, V., Martinet, P., Flament, T., Cuesta, J., Dabas, A., Olivier, M., Huber, D.: ADM-Aeolus L2A Algorithm Theoretical Baseline Document, AE-TN-IPSL-GS-001, 5.5, 89 pp., 2017.

**Specific comments:**

Some additional proofreading for English language/grammar should catch some minor errors. Remaining comments listed by page/line#. A "Fair" rating is listed under Scientific Quality because it's not quite at the "Good" level with respect to referencing related work and being clear on the issues addressed, but with minor improvements as listed above and in the following comments, it will likely be above good. Overall, this is an interesting and useful paper for the field of double-edge direct detection Doppler wind lidar systems.

1. Page 2 line 24-26: This sentence describing Aeolus is awkward. Perhaps reword as "The novel combination of these two techniques, integrated for the first time into a single wind lidar, expands the observational altitude range from the ground to the lowermost 30 km of the atmosphere."

R: revised. Please see Sect 1 page 2, line 28-30: "The novel combination of these two

techniques, integrated for the first time into a single wind lidar, expands the observational altitude range from ground to the lowermost 30 km of the atmosphere."

**Line 29: Can delete the words, "as well" from the end of the sentence since it begins with "Furthermore".**

R: revised. Please see Sect 1 page 3, line 1-3: "Furthermore, as the first high spectral resolution lidar in space (Ansmann et al., 2007; Flamant et al., 2008), ALADIN has the potential to globally monitor cloud and aerosol optical properties to contribute to climate impact studies."

2. Page 3 Line 9: Can the authors expand a little bit on the causes of "...the atmospheric and instrumental variability" for readers not familiar with the observation approach. For example, how atmospheric pressure/temperature impact the MRRC and what varies in the instrument (temperature impacting alignment? Variations in the field of view/field angles entering the etalon? Etc.?)

R: revised. Please see Sect 1 page 3, line 15-22, see also reply to major comment 1

"However, the atmospheric temperature affects the Rayleigh-Brillouin line shape, and has a direct effect on the instrument response calibration (Dabas et al., 2008). Differences in the atmospheric temperature profile of the time and location when the MRRC was obtained and the actual wind measurements are important sources of wind bias, which are especially severe in case of large temperature differences. This is the reason why it is mandatory to consider the atmospheric temperature in the Aeolus level 2B procedure to retrieve reliable winds (Dabas et al., 2008; Rennie et al., 2017). Furthermore, some experimental limitations, which will be introduced specifically in Sect. 2.1, need to be considered carefully to achieve a reliable MRRC. Overall, the atmospheric and instrumental variability coming along with a MRRC limits the reliability and repeatability of A2D instrument response calibrations."

Line 12: update to read, "It is based on an accurate theoretical model of the FPI transmission function...."

R: revised. Please see Sect 1 page 3, line 24-25: "It is based on an accurate theoretical model of the FPI transmission function and the molecular Rayleigh backscatter spectrum."

Line 28: edit to read, "Table one lists FPI-based direct detection wind lidar systems that are capable of measuring wind information...." Note that not all existing FPI systems that can be modeled this way are listed in the table, there are others in existence.

R: revised. Please see Sect 2 page 4, line 8-10: "Table 1 lists several FPI-based direct detection wind lidar systems that are capable of measuring wind information based on measurement or simulation approach."

- Page 4 Line 10 Should be "atmospheric conditions"
   R: revised. Please see Sect 2.1 page 5, line 3-4: "..., the calibration procedure can be carried out frequently based on atmospheric conditions (Dou et al., 2014; Liu et al., 2002),..."
- 4. Page 5 Line 19: fix to read, "...the transmission functions of the FPs for the atmospheric path

are slightly different compared to ..." Then please explain why this is (physics causing the differences).

R: revised. Please see Sect 2.2 page 6, line 19-24.

"However, the transmission characteristics of the FPIs for the atmospheric path are different from the transmission curves registered on the internal reference path during the instrument spectral registration because of a slightly different illumination of the beams in the respective paths due to different divergence and incidence angles on FPIs (Reitebuch et al., 2009)."

Line 24: "regardless of measurement or simulation method, any angular alignment drift will change the incidence angles on FPIs, and hence change their transmission characteristics." Technically, the FPI transmission characteristics should be a function of incidence angles, field of view, temperature, pressure, thickness or gap length, finesse, etc. so perhaps the better term here (and elsewhere) is to say that "any angular alignment drift will change in the incidence angles on the FPIS, resulting in a different transmission value." (or something similar).

R: revised. Thanks for your comments, it has been revised as "Furthermore, the FPIs transmission characteristics should be a function of incidence angles, field of view, temperature, pressure, thickness, and so forth, regardless of measurement or simulation method, any angular alignment drift will change in the incidence angles on the FPIs, resulting in a different transmission value." Please see Sect 2.2, page 6 line 30-31 to page 7 line 1-2.

5. Page 6 Line 5: This is an unusual mix of variables (wavelength and frequency shift), but ok. Line 17 and 19: The authors state that Equations 3 and 4 represent convolutions, but this is not mathematically so. These are integrations over frequency of the product of the FPI transfer function times the specific input spectrum value at that frequency. Likewise, integrating this product over only one free spectral range implies that the authors assume there are never any signals outside the etalon FSRs (e.g. where the etalon can start to transmit again). This may be practically true for most applications/wind speeds/platform pointing motions, etc. but should at least be stated as an assumption.

R: revised. Please see the updated Equation 3 and 4 in Sect 3, page 7, line 22-24.

Still we state that the equation describes the convolution of the respective functions, as we first calculated the intensity values for all  $f_i$  and thus calculate a function of the transmitted intensities depending on  $f_i$ . Afterwards this function can be used to calculate the transmitted intensity for a respective frequency  $f_i$ . Thus, mathematically, this is not only the product but indeed the convolution of the respective functions.

$$I_{A,B,INT}(f_i) = \int_{-\infty}^{+\infty} T_{A,B,INT}(f) S_i(f_i - f) df$$
$$I_{A,B,ATM}(f_a) = \int_{-\infty}^{+\infty} T_{A,B,ATM}(f) S_a(f_a - f) df$$

6. Page 7 Line 5: defects could be in the FPI mirror surface(s) (plural) right?
R: revised. Please see Sect 3, page 8 line 15-16: "..., however, small defects on the FPI mirror surfaces or of the illumination of the FPI could result in small deviations, ..."

Line 7: Why not also mention/reference the works of Spinhirne, McGill.

R: revised. Thanks for your suggestion, the related reference has been added in the revised manuscript. Please see Sect 3, page 8 line 15-16.

"However, small defects on the FPI mirror surfaces or of the illumination of the FPI could result in small deviations that have to be considered for an accurate analysis (McGill et al., 1998)."

**Reference:**

McGill, M. J., and Spinhirne, J. D.: Comparison of two direct-detection Doppler lidar techniques. Opt. Eng., 37 (10), 2675-2686, https://doi.org/10.1117/1.601804, 1998.

**Line 9: R is the mean reflectivity of the etalon mirrors? (again, plural?)**

R: revised. Please see Sect 3, page 8 line 13-14: "R is the mean reflectivity of the mirror surfaces and"

Line 14: Suggest instead to say, "An easily calculated analytical expression...." R: revised. Please see Sect 3, page 8 line 19-20: "An easily calculated analytical expression of Tenti S6 line shape model for,...."

Lines 16-21: The paper might read more easily if this paragraph was moved up earlier in the discussion.

R: revised. The sentence "The spectrally narrowband Mie scattering which is not filtered out by the Fizeau interferometer will enter the FPIs and can be considered as Mie contamination of the Rayleigh signal." has been moved to Sect 2.1 Page 5 Line 23-24. We didn't change the position of the rest of the paragraph in order to read more easily, because the mentioned variables need to be described firstly.

Line 21: The "magenta" filled area appears more "pink" – perhaps use that term instead, or "light magenta"

R: revised. It has been revised as "light magenta". Please see Sect 3 Page 9 Line 8.

7. Page 8 Line 1: Here the authors could clarify for the readers not familiar with double edge approach why the biases are worse when Mie signal is significant but not good enough to measure winds using the Mie channel. Can this be shown somehow in Figure 2?

R: it has been revised as "According to previous studies (Dabas et al., 2008), it is implied that the Mie contamination correction could improve the quality of Rayleigh wind in the cases of intermediate  $\rho$ , e.g. below 1.5, as in this case the Mie signal is not high enough to guarantee an accurate Mie wind measurement but rather becomes significant for Rayleigh channel (Sun et al., 2014; Lux et al., 2018). Please see Sect 3 Page 9 Line 14-17.

Line 4 (paragraph 2): clarify that the procedure is done assuming no Mie interference (or otherwise?)

R: revised. Please see Sect 3 Page 9 Line 29 to page 10 line 1: "It is noted that the procedure is done assuming no Mie interference."

Line 9: The text says that the red-square marks +/- 850 MHz, but the figure looks like its closer to 1 GHz. Please rectify one or the other to match.

R: Figure 3 has been updated in the revised manuscript.

Figure 3: (a) The Simulated Rayleigh Response Calibration (SRRC) for internal reference (INT, blue line) and atmospheric return (ATM, black line), the frequency of the filter cross point is marked with a red dotted line, (b) INT (blue dots) and ATM (black dots) response and corresponding linear least squares fit (blue line for INT, black line for ATM) calibration with a frequency interval of ±850 MHz, where relative frequency is used instead of absolute frequencies, (c) the non-linearities of simulated (dots) and fitted (lines) response functions from INT (blue) and ATM (black). (d) response function residuals from INT (blue line).

Line 22: clarify that the "Then the fit of the SRRC for the internal reference and atmospheric paths can be expressed as a sum of a linear fit plus a 5th order polynomial:"

R: revised. It has been revised as "Then the fit of the SRRC for the internal reference and atmospheric paths can be expressed as a sum of a linear fit and a 5th order polynomial fit, that is," Please see Sect 3 Page 10 Line 16-17.

Page 9 Line 5: replace "In the frame of..." with "As part of.." The rest of this paragraph would benefit from additional proofreading for English grammar.
 R: revised. Please see Sect 4 Page 11 Line 2: "As part of the North Atlantic Waveguide and Downstream Experiment (NAWDEX) carried out in, ..."

Line 19-20: Suggest a rewrite to read "Time-space matching datasets between dropsonde and A2D can be used as both references to validated A2D wind measurements and to provide essential...."

R: revised, it has been revised as "Time-space matching datasets between dropsonde and A2D can be used as both references to validate A2D wind measurements and to provide essential atmospheric temperature and pressure profiles for SRRC in this study." Please see Sect 4 Page 11 Line 16-18.

Line 23-24: This sentence repeats a little bit of what was written before, now referring to "illumination properties" - can you be more specific? Is this a function of differences in the

**spatial (e.g. the pupil) distribution or in the field (e.g. angular) distribution?**

R: revised. It has been revised as "It is noted that the transmission functions of the FPIs are reproducible, and the transmission characteristics are different for the internal reference and atmospheric path due to the difference of the illumination of the FPIs in these two paths. This refers to the difference of divergence and incidence angles on the FPIs for the respective paths. It is both a difference in the spatial as well as in the angular distribution of the light. In particular, the use of a multimode fibre in the internal reference path gives rise to speckles, resulting in an intensity distribution which is markedly different from that of the atmospheric path." Please see **Sect 4 Page 11 Line 21-26.**

9. Page 10 Line 13-14: The authors state that the, "measured response values obtained from A2D wind velocity measurement mode are brought into the fitted SRRC...." What does "brought into" mean here? Is this a mapping? What is the process for doing this? R: revised. Change "brought into..." to "combined with...", the specific process for doing this

is marked with red-line square in the figure below. Please see Sect 5 Page 12 Line 15.

---

## Author Response (AR1)

Dear Editor and referees,

Thank you for your review of our manuscript. We greatly appreciate the substantial amount of time and effort that you dedicated to this review process.

We have revised the manuscript according to your comments and the point-by-point responses are attached in this file. The marked-up manuscript version showing the changes made is also provided as follow.

**Response to Referee Comment 1 on** "Rayleigh wind retrieval for the ALADIN airborne demonstrator of the Aeolus mission using simulated response calibration"**

**Comments:**

1. The paper shows how calibration curves for a direct-detection airborne Doppler lidar can be derived from the known pressure and temperature in the sensed atmospheric volume and a careful characterization of the transmission characteristics of the interferometers used in the receiver. The calibration procedure is a copy from what is done for AEOLUS. It is shown that the procedure can be applied as well to the airborne demonstrator of AEOLUS, and achieves a better accuracy with a reduced bias and equivalent standard deviation with collocated drop-sonde wind measurements as with a measured response curve that does not take specifically into account the pressure and temperature conditions.

R: Thanks for your comment. It should be noted that the A2D SRRC procedure mentioned in this paper is not a pure "copy" from what is done for ALADIN. There are some significant differences, especially in the generation and update of the transmission characteristics of the FPIs of the Rayleigh receiver for the atmospheric channel. The specific differences are listed below:

1. The transmission characteristics of FPIs for the atmospheric path are different from the transmission curves registered on the internal reference path during the instrument spectral registration because of the difference of the illumination of the beams in the atmospheric and the internal reference paths due to different divergence and incidence angles on FPIs (Reitebuch et al., 2009). As opposed to ALADIN, where only the transmission curve in the internal reference path can be measured during instrument spectral registration, the A2D FPI transmission curves both in the internal reference path and in the atmospheric path were measured in previous campaigns, demonstrating slight deviations between both transmission paths due to the aforementioned reasons. Therefore, different combinations of FPI transmission functions derived from different campaigns can be used to derive different candidate SRRCs. After the comparison of candidate SRRCs with simultaneous MRRC, the most satisfactory combination is used for initial SRRC determination.

2. As for ALADIN, the core idea of the updated spectral registration using the Airy and top-hat function is based on the comparison of the predicted one and an MRRC. The FPIs transmission characteristics cannot represent the actual sensitivity of the Rayleigh receiver at the atmospheric

path until the difference of predicted and the measured responses coincide within a threshold limit. But for A2D, the optical path characteristic of the A2D Rayleigh channel is considered carefully. Basically, the FPI center frequency is sensitive to the incidence angle of the light. It is a reasonable way to optimize the FPI transmission function by fine adjusting the center frequency of filter A or B for the atmospheric path. The Rayleigh spectrometer is composed of two FPIs which are sequentially coupled. Thus, the reflection of the directly illuminated FPI is directed to the second FPI. Any incidence angle change in front of the Rayleigh spectrometer will act similarly on both FPIs. The related description has been added in Sect 7 Page 21 Line 5-18.

2. The practical significance of the method should be discussed. The paper suggests the transmission characteristics of the two FPs are very stable, except for a frequency shift caused by an incidence angle varying from one flight to the other. In Fig 10 or 13, the results are obtained with a frequency shift determined from data acquired during the same flight. Will the frequency shift be significantly modified during another flight? If yes, this should be stressed and a conclusion should be that response calibration should be done every flight.

R: revised. Thank you for this comment. We will add a clarification to the manuscript.

The derived frequency shift of 20 MHz can basically depend on the alignment of the atmospheric optical path. From the experience from the last 10 years it is known that this alignment is not randomly varying from flight to flight, but changes from campaign to campaign. As the telescope and optical receiver is coupled via free optical path (and not via a fibre), the mechanical integration of the A2D into the aircraft prior to each campaign leads to small variation in position and incidence angle on the spectrometers for each deployment. Thus, a valid response calibration can be used for the entire campaigns period. This is true for both, measured or rather simulated response calibrations. In order to monitor the atmospheric path alignment, the position of the spots generated on the ACCD detector behind each FPI is analyzed and serves as information on the alignment during the flight itself and among the flights during the campaigns period. It should be noted that the applied frequency shift is only 20 MHz, which is even less than the frequency separation of successive measurement points during a response calibration (25 MHz) and which corresponds to  $1.8 \times 10^3$  of the FSR of the FPIs. The related description has been added to **Sect 5.3 Page 17 Line 14-23**.

**3. The SRRC reduced the bias, but on the other hand lower the correlation coefficient with dropsonde vlos in Fig 10. This should be commented.**

R: revised. The comparison of the correlation coefficient has been added in Sect 6 Page 18 line 5-18: "The correlation coefficient r, bias and standard deviation are also calculated and listed in Table 5. Fig. 10 (a) illustrates the comparison of LOS wind velocity between dropsonde and A2D Rayleigh channel measurement, showing that the fit parameters slightly deviate from the ideal case. The correlation coefficient r, bias and standard deviation of the A2D Rayleigh winds are 0.95, 0.23 m s-1 and 2.20 m s-1, respectively, which is comparable to results in previous studies (Lux et al., 2018). The comparison of LOS wind velocity between dropsonde measurements and the results derived from SRRC without FPIs optimization is illustrated in Fig. 10 (b). The corresponding correlation coefficient r, bias and standard deviation are determined to be 0.93, -3.32 m s-1 and 2.61 m s-1, respectively. It can be seen that underestimation of the LOS wind velocity from SRRC without the FPIs optimization is significant, demonstrating the necessity of the FPIs optimization before wind retrieval when using SRRC procedure. Figure 10 (c) shows the comparison of LOS wind velocity between dropsonde measurements and results derived from SRRC with FPIs optimization. The bias is 0.05 m s-1, which is better than the results from A2D wind with MRRC, and the correlation coefficient r and standard deviation are 0.94 and 2.52 m s-1, respectively. This is comparable to the results from A2D Rayleigh channel measurements and implies the feasibility and robustness of SRRC with FPIs optimization on A2D Rayleigh wind retrieval. From now on, only SRRC results with optimized FPI parameters will be discussed."

4. The paper mentions the presence of an internal reference channel without explaining exactly what it is. A simple graph showing the internal reference and the atmospheric path would improve the clarity of the paper.

R: revised. Thanks for your suggestion, we didn't explain it clearly. The specific schematic of ALADIN Airborne Demonstrator (A2D) was shown in Fig. 1 in (Lux et al., 2018), which has been already referenced in the following added in Sect 2, page 4, line 16-31 see below:

"For each direct detection wind lidar system, the emitted laser frequency should be known in order to allow an accurate derivation of the Doppler frequency shift. A zero Doppler shift reference determined by pointing to the zenith direction has been used to correct the short-term frequency drift in previous studies (Souprayen et al., 1999b; Korb et al., 1992; Dou et al., 2014). But for the A2D, the internal reference path is particularly dedicated to the derivation of information about the emitted laser frequency. As shown in Lux et al. (2018, Fig. 1), a small portion of the laser beam radiation is collected by an integrating sphere and coupled into a multi-mode fibre, then injected into the receiver via the front optics. This path is called internal reference path. The atmospheric backscattered signal is collected by a Cassegrain telescope and guided via free optical path propagation to the front optics and receiver successively. This path is called the atmospheric path. An electro-optical modulator is used to temporally separate the atmospheric signal from the internal reference signal, thereby avoiding disturbances of the internal reference signal by atmospheric signal and saturation of the detectors at short ranges (Reitebuch et al., 2009). Because of the different optical illumination of the internal and atmospheric path resulting in different divergence and incidence angles on the FPIs, the response calibration curves for these two paths are different. It is noted that the internal reference path of ALADIN is different from A2D's, where ALADIN uses free path propagation rather than fibre coupling unit."

The related descriptions of the internal reference path and atmospheric path are also updated:

1. Sect 2.1, page 5, line 16-19, "For ALADIN, the Rayleigh winds produced by the level 1B processor (Reitebuch et al., 2018) are based on a MRRC while the level 2B processor uses a

SRRC. A MRRC includes three response calibration curves, one each derived from the internal reference, the atmospheric and the ground return."

- In Sect 2.2, page 7, line 2-4, "Regarding the A2D, a SRRC based on such a simulation approach promises an improvement in terms of wind speed errors. A SRRC includes two response calibration curves derived from internal reference path and atmospheric path."
- 5. Page 2, line 9: in the CDL, the backscattered light captured by the telescope is mixed with a frequency shifted emitter laser. The frequency shift enables the measurement of positive and negative winds. It is not mentioned.

R: revised. Please see Sect 1, page 2, line 12-13: "..., light, and the frequency shift introduced by an acoustic-optical modulator enables the measurement of positive and negative winds."

6. Equations 3 and 4: there integrals should be between -\infty and +\infty. In practice S\_a has a limited width so the limits -FSR/2 +FSR/2 can be enough if FSR is much larger, but +-\infty is better.
R: revised. Please see the updated Equation 3 and 4 in Sect 3, page 8, line 2-4.

$$I_{A,B,INT}(f_i) = \int_{-\infty}^{+\infty} T_{A,B,INT}(f) S_i(f_i - f) df$$
$$I_{A,B,ATM}(f_a) = \int_{-\infty}^{+\infty} T_{A,B,ATM}(f) S_a(f_a - f) df$$

7. Page 11, lines 13-20: it is suggested the atmospheric and internal characteristics of FP transmissions are solely due to plate defects. This is wrong. The main reason is the beam étendue is different in the two channels due to a diaphragm.

R: revised. Thanks for your comment, yes, we didn't explain it correctly at this point. It has been revised as "the transmission functions of FPIs for the atmospheric path are different from the transmission curves registered on the internal reference path during the instrument spectral registration. This is because of the difference in the illumination of the FPIs by the beams in the atmospheric and the internal reference paths, i.e. due to different divergence and incidence angles (Reitebuch et al., 2009)." Please see Sect 2.2, page 6, line 28-31.

Page 12, lines 19-23: the authors should write what \eplison\_R is. It is the difference between the SRRC and the MRRC. Ideally it should be randomly fluctuations about 0 with no offset not slope.
 R: revised. The definition of εR has been updated in the revised manuscript as

" $\varepsilon_R$  is defined as the difference between response from the respective SRRCs and the MRRC. Then, the linear fit of  $\varepsilon_R$  as function of f' is made, returning a slope  $\varepsilon_{R\_slope}$  and intercept  $\varepsilon_{R\_intercept}$ based on Eqs. (18A) – (18B) in (Dabas and Huber, 2017). Ideally, if the results from the SRRC and MRRC match,  $\varepsilon_R$  should be randomly fluctuations about 0 with zero  $\varepsilon_{R\_intercept}$  and  $\varepsilon_{R\_slope}$ .", please see Sect 5.2, page 15, line20-24.

**Response to Referee Comment 2 on "Rayleigh wind retrieval for the ALADIN airborne demonstrator of the Aeolus mission using simulated response calibration"**

**Major comments:**

This paper presents an alternative technique for retrieving LOS wind estimates from the molecular channel of the Aeolus Airborne Demonstrator (A2D) using modeled response functions ("Simulated Rayleigh Response Calibration" or SRRC) derived using best-fit instrument models and the given atmospheric conditions (temperature and pressure) when available from other observations.

 The SRRC approach provides some advantages over the "traditional" double-edge approach of measuring calibration response curves during the test process (the MRRC approach), but the authors could do a better job of explaining the reasoning behind this (vs. just listing numbers) at the beginning of the paper and in the abstract. The approach is a good idea, especially when faced with consistent Mie contamination during flight tests.

R: revised.

In the abstract, page 1, line 14-21, the reason why SRRC provides advantages over MRRC is added and revised as ".... However, differences exist between the respective atmospheric temperature profiles that are present during the conduction of the MRRC and the actual wind measurements. These differences are an important source of wind bias since the atmospheric temperature has a direct effect on the instrument response calibration. Furthermore, some experimental limitations and requirements need to be considered carefully to achieve a reliable MRRC. The atmospheric and instrumental variability thus currently limit the reliability and repeatability of a MRRC. In this paper, a procedure for a simulated Rayleigh response calibration (SRRC) is developed and presented in order to resolve these limitations of the A2D MRRC."

In addition, related introductions are also added in Sect.1, Page 3, line 14-24, "Currently, only measured Rayleigh response calibrations (MRRC) are used for the A2D (Marksteiner, 2013; Lux et al., 2018; Marksteiner et al., 2018). However, the atmospheric temperature affects the Rayleigh-Brillouin line shape and has a direct effect on the instrument response calibration (Dabas et al., 2008). Differences exist between the respective atmospheric temperature profiles that are present during the conduction of the MRRC and the actual wind measurements. These differences are an important source of wind bias which grows with increasing temperature in the Aeolus level 2B procedure to retrieve reliable winds (Dabas et al., 2008; Rennie et al., 2017). Furthermore, some experimental limitations, which will be introduced specifically in Sect. 2.1, need to be considered carefully to achieve a reliable MRRC. Overall, the atmospheric and instrumental variability coming along with a MRRC limits the reliability and repeatability of A2D instrument response calibrations."

Have other double-edge wind lidar researchers done anything similar to this before?

R: Yes, as shown in Table 1, there are several FPI-based direct detection wind lidar systems that are capable of measuring wind based on a measurement approach or a simulation approach. The black-marked parts use a simulation approach to obtain calibration response curves, which is similar to the SRRC method mentioned in this paper.

| Lidar                                      | Wavelength and system                             | Calibration
approach                                                                   | Instrument
drift
correction              | References                                                                               |
|--------------------------------------------|---------------------------------------------------|-------------------------------------------------------------------------------------------|------------------------------------------------|------------------------------------------------------------------------------------------|
| OHP a
Rayleigh lidar         | 532 nm, double
FPIs                            | Simulation,
FPI scanning                                                               | quick wind
acquisition
cycle strategy    | Chanin et al., 1989;
Garnier and Chanin,
1992; Souprayen et al.,
1999a, 1999b   |
| NASA b
Rayleigh/Mie
lidar | 355 nm, three
FPIs                             | Simulation
FPI or laser
scanning                                                    | locking etalon
and servo-
control system | Korb et al., 1992; Korb
et al. 1998; Flesia and
Korb, 1999; Flesia et
al., 2000 |
| USTC c
Rayleigh lidar        | 355 nm, three
FPIs                             | measurement and simulation, FPI scanning                                                  | locking etalon
and servo-
control system | Xia et al., 2012;
Dou et al., 2014                                                    |
| ESA
ALADIN                              | 355 nm, double
FPIs for
Rayleigh
channel | level 1B:
measurement,
laser scanning
level 2B:
simulation,
laser scanning | internal
reference path                     | Reitebuch et al., 2018;
Rennie et al., 2017                                           |
| DLR
A2D                                 | 355 nm, double
FPIs for
Rayleigh
channel | Measurement,
laser scanning                                                            | internal
reference path                     | Marksteiner, 2013; Lux
et al.,2018;
Marksteiner et al., 2018                       |

**Table 1. Comparison of different FPI-based direct detection wind lidars**

a Observatory of Haute Provence, France

b National Aeronautics and Space Administration, U.S.

c University of Science and Technology of China, China. This lidar is mobile.

2. An explanation of the physical differences between the internal reference channel and the atmospheric channel would be helpful. For example, does the IRC have a different set of field angles into the FP etalons than provided by the telescope/receive path returns? Does the IRC only see narrowband light?

R: revised. Thanks for your suggestion, we didn't explain it clearly. Yes, the atmospheric path and internal reference path differ in their field angles. The internal reference signal is coupled into the receiver via an optical fiber whereas the atmospheric signal enters the receiver via free beam bath through a set of different optics. This leads to a slightly different set of field angles on the FPIs for the internal path and the atmospheric path. During the ISR only the internal path, illuminated with spectrally narrow-band light from the laser is recorded, while for the IRC the internal path (with narrow spectral bandwidth from the laser) and the atmospheric path with broad spectral bandwidth molecular returns, but also narrow spectral bandwidth cloud, aerosol and ground returns is recorded.

The specific schematic of ALADIN Airborne Demonstrator (A2D) was shown in Fig. 1 in (Lux et al., 2018), which has been already referenced in the following added paragraph in Sect 2, page 4, line 16-31, see below:

"For each direct detection wind lidar system, the emitted laser frequency should be known in order to allow an accurate derivation of the Doppler frequency shift. A zero Doppler shift reference determined by pointing to the zenith direction has been used to correct the short-term frequency drift in previous studies (Souprayen et al., 1999b; Korb et al., 1992; Dou et al., 2014). But for the A2D, the internal reference path is particularly dedicated to the derivation of information about the emitted laser frequency. As shown in Lux et al. (2018, Fig. 1), a small portion of the laser beam radiation is collected by an integrating sphere and coupled into a multi-mode fibre, then injected into the receiver via the front optics. This path is called internal reference path. The atmospheric backscattered signal is collected by a Cassegrain telescope and guided via free optical path propagation to the front optics and receiver successively. This path is called the atmospheric path. An electro-optical modulator is used to temporally separate the atmospheric signal from the internal reference signal, thereby avoiding disturbances of the internal reference signal by atmospheric signal and saturation of the detectors at short ranges (Reitebuch et al., 2009). Because of the different optical illumination of the internal and atmospheric path resulting in different divergence and incidence angles on the FPIs, the response calibration curves for these two paths are different. It is noted that the internal reference path of ALADIN is different from A2D's, where ALADIN uses free path propagation rather than fibre coupling unit."

The related descriptions of the internal reference path and atmospheric path are also updated:

- In Sect 2.1, page 5, line 16-19, "For ALADIN, the Rayleigh winds produced by the level 1B processor (Reitebuch et al., 2018) are based on a MRRC while the level 2B processor uses a SRRC. A MRRC includes three response calibration curves, one each derived from the internal reference, the atmospheric and the ground return."
- 2. In Sect 2.2, page 6, line 25-28, "Regarding the A2D, a SRRC based on such a simulation approach promises an improvement in terms of wind speed errors. A SRRC includes two response calibration curves derived from internal reference path and atmospheric path."
- 3. The paper would also benefit from a short, clear discussion on the topic of Mie (aerosol)

contamination on the Rayleigh calibration as the topic comes up several times in the paper. Present the reasons for the aerosol induced bias and reference the literature. This could be followed by cleaning up some paragraphs that vaguely refer the issue, without explaining it.

R: revised. Thanks for your suggestion. About the topic of Mie contamination on Rayleigh calibration, we have updated related paragraphs.

Firstly, in Sect 2.1, page 5, line 31 to page 6, line 1-6, we discuss the reasons for the aerosol induced bias: "Firstly, the particulate Mie scattering which is not fully filtered out by the Fizeau interferometer will enter the FPIs and can be considered as Mie contamination of the Rayleigh signal. Because of the different spectral widths of the particle and molecular backscatter signal, the sensitivities of the FPIs on them are different. If not taken into account, the Mie contamination on the Rayleigh channel is one of the sources of systematic errors because it modifies the MRRC curve. In order to avoid such modifications, the A2D tries to conduct IRCs in preferably pure Rayleigh atmosphere."

Then, we analyze the LOS wind velocity error induced by Mie contamination in Sect 3 page 9 line 19-25 based on simulation results: "The LOS wind velocity error  $\Delta V_{MC}$  induced by Mie contamination is defined as the difference of the LOS wind velocities measured under purely atmospheric molecular conditions and conditions with a scattering ratio of  $\rho$ . Figure 2 shows a simulation of  $\Delta V_{MC}$  at T=223 K and P=301 hPa, where the x-axis and y-axis represent different response values and scattering ratios, respectively. Positive and negative  $\Delta V_{MC}$  represent the overestimation and underestimation of the LOS velocity, respectively. An overestimation of LOS velocities occurs at response values less than 0.235 in this case. Larger scattering ratios result in larger overestimation, and the difference can get up to 13 m s-1 in case of  $\rho = 3$ ."

We also introduce the effect of Mie contamination correction on systematic error optimization in Sect 3 page 9 line 25-27 to page 10 line 1-6: "According to previous studies (Dabas et al., 2008), the Mie contamination correction could improve the quality of Rayleigh winds in cases of intermediate  $\rho$ , e.g. below 1.5. In this region the Mie signal is not high enough to guarantee an accurate Mie wind measurement but instead becomes rather significant for the Rayleigh channel (Sun et al., 2014; Lux et al., 2018). The value of  $\rho$ , which is needed for the Mie contamination correction in the Rayleigh channel, is obtained by analysing the Mie channel signal. The detailed algorithm can be seen in (Flamant et al., 2017). " Reference:

 Flamant, P., Lever, V., Martinet, P., Flament, T., Cuesta, J., Dabas, A., Olivier, M., Huber, D.: ADM-Aeolus L2A Algorithm Theoretical Baseline Document, AE-TN-IPSL-GS-001, 5.5, 89 pp., 2017.

**Specific comments:**

Some additional proofreading for English language/grammar should catch some minor errors. Remaining comments listed by page/line#. A "Fair" rating is listed under Scientific Quality because it's not quite at the "Good" level with respect to referencing related work and being clear on the issues addressed, but with minor improvements as listed above and in the following comments, it will likely be above good. Overall, this is an interesting and useful paper for the field of double-edge direct detection Doppler wind lidar systems.

Page 2 line 24-26: This sentence describing Aeolus is awkward. Perhaps reword as "The novel combination of these two techniques, integrated for the first time into a single wind lidar, expands the observational altitude range from the ground to the lowermost 30 km of the atmosphere."
 R: revised. Please see Sect 1 page 2, line 27-29: "The novel combination of these two techniques, integrated for the first time into a single wind lidar, expands the observable altitude range from ground to the lowermost 30 km of the atmosphere."

Line 29: Can delete the words, "as well" from the end of the sentence since it begins with "Furthermore".

R: revised. Please see Sect 1 page 3, line 1-3: "Furthermore, as the first high spectral resolution lidar in space (Ansmann et al., 2007; Flamant et al., 2008), ALADIN has the potential to globally monitor cloud and aerosol optical properties to contribute to the climate impact studies."

Page 3 Line 9: Can the authors expand a little bit on the causes of "...the atmospheric and instrumental variability" for readers not familiar with the observation approach. For example, how atmospheric pressure/temperature impact the MRRC and what varies in the instrument (temperature impacting alignment? Variations in the field of view/field angles entering the etalon? Etc.?)
 R: revised. Please see Sect 1 page 3, line 15-24, see also reply to major comment 1

"However, the atmospheric temperature affects the Rayleigh-Brillouin line shape and has a direct effect on the instrument response calibration (Dabas et al., 2008). Differences exist between the respective atmospheric temperature profiles that are present during the conduction of the MRRC and the actual wind measurements. These differences are an important source of wind bias which grows with increasing temperature differences. This is also the reason why it is mandatory to consider the atmospheric temperature in the Aeolus level 2B procedure to retrieve reliable winds (Dabas et al., 2008; Rennie et al., 2017). Furthermore, some experimental limitations, which will be introduced specifically in Sect. 2.1, need to be considered carefully to achieve a reliable MRRC. Overall, the atmospheric and instrumental variability coming along with a MRRC limits the reliability and repeatability of A2D instrument response calibrations."

Line 12: update to read, "It is based on an accurate theoretical model of the FPI transmission function...."

R: revised. Please see Sect 1 page 3, line 26-27: "It is based on an accurate theoretical model of the FPI transmission function and the molecular Rayleigh backscatter spectrum."

Line 28: edit to read, "Table one lists FPI-based direct detection wind lidar systems that are capable of measuring wind information...." Note that not all existing FPI systems that can be modeled this way are listed in the table, there are others in existence.

R: revised. Please see Sect 2 page 4, line 12-14: "Table 1 lists several FPI-based direct detection wind lidar systems that are capable of measuring wind information based on a measurement approach or a simulation approach."

3. Page 4 Line 10 - Should be "atmospheric conditions"

R: revised. Please see Sect 2.1 page 5, line 9-11: "Regarding ground-based lidar systems, the calibration procedure can be carried out frequently. Based on stable atmospheric conditions (Dou et al., 2014; Liu et al., 2002),..."

Page 5 Line 19: fix to read, "...the transmission functions of the FPs for the atmospheric path are slightly different compared to ..." Then please explain why this is (physics causing the differences).
 R: revised. Please see Sect 2.2 page 6, line 26-31.

"However, the transmission functions of FPIs for the atmospheric path are different from the transmission curves registered on the internal reference path during the instrument spectral registration. This is because of the difference in the illumination of the FPIs by the beams in the atmospheric and the internal reference paths, i.e. due to different divergence and incidence angles (Reitebuch et al., 2009)."

Line 24: "regardless of measurement or simulation method, any angular alignment drift will change the incidence angles on FPIs, and hence change their transmission characteristics." Technically, the FPI transmission characteristics should be a function of incidence angles, field of view, temperature, pressure, thickness or gap length, finesse, etc. so perhaps the better term here (and elsewhere) is to say that "any angular alignment drift will change in the incidence angles on the FPIS, resulting in a different transmission value." (or something similar).

R: revised. Thanks for your comments, it has been revised as "Furthermore, FPI transmission functions should be a function of incidence angles, field of view, temperature, pressure, thickness, fitness and so forth. Regardless of measurement or simulation method, any angular alignment drift will change the incidence angles on the FPIs, resulting in a different transmission value." Please see **Sect 2.2, page 7 line 7-10**.

5. Page 6 Line 5: This is an unusual mix of variables (wavelength and frequency shift), but ok. Line 17 and 19: The authors state that Equations 3 and 4 represent convolutions, but this is not

mathematically so. These are integrations over frequency of the product of the FPI transfer function times the specific input spectrum value at that frequency. Likewise, integrating this product over only one free spectral range implies that the authors assume there are never any signals outside the etalon FSRs (e.g. where the etalon can start to transmit again). This may be practically true for most applications/wind speeds/platform pointing motions, etc. but should at least be stated as an assumption.

R: revised. Please see the updated Equation 3 and 4 in Sect 3, page 8, line 1-4.

Still we state that the equation describes the convolution of the respective functions, as we first calculated the intensity values for all  $f_i$  and thus calculate a function of the transmitted intensities depending on  $f_i$ . Afterwards this function can be used to calculate the transmitted intensity for a respective frequency  $f_i$ . Thus, mathematically, this is not only the product but indeed the convolution of the respective functions.

$$I_{A,B,INT}(f_i) = \int_{-\infty}^{+\infty} T_{A,B,INT}(f) S_i(f_i - f) df$$
$$I_{A,B,ATM}(f_a) = \int_{-\infty}^{+\infty} T_{A,B,ATM}(f) S_a(f_a - f) df$$

6. Page 7 Line 5: defects could be in the FPI mirror surface(s) (plural) right?
R: revised. Please see Sect 3, page 8 line 21-22: "However, small defects on the FPI mirror surfaces or imperfect illumination of the FPI could result in small deviations that have to be considered (McGill et al., 1998).

**Line 7: Why not also mention/reference the works of Spinhirne, McGill.**

R: revised. Thanks for your suggestion, the related reference has been added in the revised manuscript. Please see Sect 3, page 8 line 21-22.

"However, small defects on the FPI mirror surfaces or imperfect illumination of the FPI could result in small deviations that have to be considered (McGill et al., 1998).

Reference:

McGill, M. J., and Spinhirne, J. D.: Comparison of two direct-detection Doppler lidar techniques. Opt. Eng., 37 (10), 2675-2686, https://doi.org/10.1117/1.601804, 1998.

**Line 9: R is the mean reflectivity of the etalon mirrors? (again, plural?)**

R: revised. Please see Sect 3, page 9 line 3: "R is the mean reflectivity of the mirror surfaces and,..."

**Line 14: Suggest instead to say, "An easily calculated analytical expression...."**

R: revised. Please see Sect 3, page 9 line 8-10: "An easily calculated analytical expression of the Tenti S6 line shape model for atmospherically relevant temperatures and pressures is used herein (Witschas, 2011a, b; Witschas et al., 2014)."

Lines 16-21: The paper might read more easily if this paragraph was moved up earlier in the discussion.

R: revised. The sentence "the particulate Mie scattering which is not fully filtered out by the Fizeau interferometer will enter the FPIs and can be considered as Mie contamination of the Rayleigh signal." has been moved to Sect 2.1 Page 5 Line 31 to page 6 line 1. We didn't change the position of the rest of the paragraph in order to read more easily, because the mentioned variables need to be described firstly.

Line 21: The "magenta" filled area appears more "pink" – perhaps use that term instead, or "light magenta"

R: revised. It has been revised as "light magenta". Please see Sect 3 Page 9 Line 17.

7. Page 8 Line 1: Here the authors could clarify for the readers not familiar with double edge approach why the biases are worse when Mie signal is significant but not good enough to measure winds using the Mie channel. Can this be shown somehow in Figure 2?

R: it has been revised as "According to previous studies (Dabas et al., 2008), the Mie contamination correction could improve the quality of Rayleigh winds in cases of intermediate  $\rho$ , e.g. below 1.5. In this region the Mie signal is not high enough to guarantee an accurate Mie wind measurement but instead becomes rather significant for the Rayleigh channel (Sun et al., 2014; Lux et al., 2018)." Please see Sect 3 Page 9 Line 25-27 to Page 10 Line 1-2.

Line 4 (paragraph 2): clarify that the procedure is done assuming no Mie interference (or otherwise?) R: revised. Please see Sect 3 Page 10 Line 9-10: "It is noted that the procedure is done assuming no Mie contamination in this case."

Line 9: The text says that the red-square marks +/- 850 MHz, but the figure looks like its closer to 1 GHz. Please rectify one or the other to match.

R: Figure 3 has been updated in the revised manuscript.

Figure 3: (a) The Simulated Rayleigh Response Calibration (SRRC) for internal reference (INT, blue line) and atmospheric return (ATM, black line), the frequency of the filter cross point is marked with a red dotted line, (b) INT (blue dots) and ATM (black dots) response and corresponding linear least squares fit (blue line for INT, black line for ATM) calibration with a frequency interval of ±850 MHz, where relative frequency is used instead of absolute frequencies, (c) the non-linearities of simulated (dots) and fitted (lines) response functions from INT (blue) and ATM (black). (d) response function residuals from INT (blue line) and ATM (black line).

Line 22: clarify that the "Then the fit of the SRRC for the internal reference and atmospheric paths can be expressed as a sum of a linear fit plus a 5th order polynomial:"

R: revised. It has been revised as "A fit of the SRRC for the internal reference and atmospheric paths can be expressed as a sum of a linear fit and a 5th order polynomial fit:" Please see Sect 3 Page 11 Line 2-3.

Page 9 Line 5: replace "In the frame of..." with "As part of.." The rest of this paragraph would benefit from additional proofreading for English grammar.
 R: revised. Please see Sect 4 Page 11 Line 12-13: "As part of the North Atlantic Waveguide and Downstream Experiment (NAWDEX) carried out in, ..."

Line 19-20: Suggest a rewrite to read "Time-space matching datasets between dropsonde and A2D can be used as both references to validated A2D wind measurements and to provide essential...." R: revised, it has been revised as "Time-space matching datasets between dropsonde and A2D can be used as both references to validate A2D wind measurements and to provide essential atmospheric temperature and pressure profiles for SRRC in this study." Please see Sect 4 Page 11 Line 27-28 to page 12 line 1.

Line 23-24: This sentence repeats a little bit of what was written before, now referring to "illumination properties" - can you be more specific? Is this a function of differences in the spatial (e.g. the pupil) distribution or in the field (e.g. angular) distribution?

R: revised. It has been revised as "The transmission functions of the FPIs are reproducible, and the transmission characteristics are different for the internal reference and atmospheric path. The underlying difference in illumination includes both a difference in the spatial as well as in the angular distribution of the light. In particular, the use of a multimode fibre in the internal reference path gives rise to speckles, resulting in an intensity distribution which is markedly different from that of atmospheric path." Please see Sect 4 Page 12 Line 4-9.

9. Page 10 Line 13-14: The authors state that the, "measured response values obtained from A2D wind velocity measurement mode are brought into the fitted SRRC...." What does "brought into" mean here? Is this a mapping? What is the process for doing this?

R: revised. Change "brought into…" to "combined with…", the specific process for doing this is marked with red-line square in the figure below. Please see Sect 5 Page 13 Line 1.